# The activating receptor NKG2D is an anti-fungal pattern recognition receptor

Yoav Charpak-Amikam [1,6], Mark Kournos[1,6], Rebecca Kotzur[1], Batya Isaacson[1], Tal Bagad Brenner [1], Elidet Gomez-Cesar[1], Ammar Abou-Kandil[2], Ronen Ben-Ami[3], Maya Korem[4], Nadia Guerra [5], Nir Osherov [2] & Ofer Mandelboim [1] ✉

NKG2D is a central activating receptor involved in target recognition and killing by Natural Killer and CD8+ T cells. The known role of NKG2D is to recognize a family of self-induced stress ligands that are upregulated on stressed cells such as cancerous or virally infected cells. Fungal pathogens are a major threat to human health, infecting more than a billion patients yearly and becoming more common and drug resistant. Here we show that NKG2D plays a critical role in the immune response against fungal infections. NKG2D can recognize fungal pathogens from most major families including *Candida*, *Cryptococcus* and *Aspergillus* species, and mice lacking NKG2D are extremely sensitive to fungal infections in models of both invasive and mucosal infections, making NKG2D an anti-fungal pattern recognition receptor.

The kingdom of fungi constitutes a major part of the human environment. Various fungal species play a central and double role in human health as being both commensals, as part of the human microbiome, and dangerous pathogens. More than 1 billion people suffer from fungal infections every year, over 100 million suffer from mucosal fungal infections and several million struggles with severe invasive or chronic infections that can have mortality rates of up to 50%[1,2].

This high burden of infection leads to more than 1.6 million deaths per year, which is greater than mortality due to malaria infections and equal to the estimated burden of tuberculosis[1,2]. In addition, multiple fungal species are also found in various cancers, affecting the immune response, with possible clinical implications on disease progression and response to various treatments[3].

The threat posed by pathogenic fungi was recently emphasized during the COVID-19 pandemic. Afflicted patients, especially those admitted to Intensive Care Units, were highly susceptible to serious fungal pathogens such as *Candida*, *Aspergillus*, and *Mucorales* members[4–7]. These infections contributed significantly to mortality, with more than 50% mortality of co-infected patients[5–8].

Indeed, the immune response to commensal and pathogenic fungi is critical to human health. A well-known portion of this immune response is mediated by myeloid cells, but lymphocytes also play a central role. This includes both adaptive responses mediated by $T_H1$, $T_H17$, and cytotoxic T cells and innate responses mediated by various Innate Lymphoid Cells (ILCs), most notably Natural Killer (NK) cells[9,10].

NK cells exert their anti-fungal response by directly killing fungal cells using lytic granules or by secreting various cytokines and chemokines that recruit additional immune cells, orchestrating the immune response to the fungal invasion[11].

The first stage in priming an immune response is the recognition of the upcoming threat, usually via pattern recognition receptors (PRRs). While many anti-fungal PRRs expressed on myeloid cells are well characterized, the PRRs used by lymphocytes are not. Some receptors expressed on T cells that have been shown to recognize fungi are CR3, CD5, and NLRP-3, and some populations of T cells were also shown to express the archetypical fungal-recognizing receptor Dectin-1[10,12]. How NK cells recognize fungi is even less known. Unlike myeloid anti-fungal PRRs, NK cell receptors mostly recognize only specific

[1]The Concern Foundation Laboratories at the Lautenberg Center for Immunology and Cancer Research, Hebrew University Medical School, IMRIC, Jerusalem, Israel. [2]Department of Clinical Microbiology and Immunology, Sackler School of Medicine, Tel-Aviv University, Ramat-Aviv, Tel-Aviv, Israel. [3]Infectious Diseases Unit, Tel Aviv Sourasky Medical Center, and the Sackler Faculty of Medicine, Tel Aviv University, Tel Aviv, Israel. [4]Department of Clinical Microbiology and Infectious Diseases, Hadassah-Hebrew University Medical Center, Jerusalem, Israel. [5]Department of Life Sciences, Imperial College London, London, UK. [6]These authors contributed equally: Yoav Charpak-Amikam, Mark Kournos. ✉e-mail: oferm@ekmd.huji.ac.il

fungi[11]. Notable examples include NKp30[13,14], NKp46[15], CD56[16,17], or the Fc-receptor CD16, which recognizes antibody-coated fungal cells[18].

In addition, we have recently reported that the immune checkpoint receptor TIGIT recognizes *C. albicans*, leading to inhibition of NK and T cell killing, and proposed that *Candida* engage TIGIT to evade immune recognition and elimination[19].

One major family of fungi-recognizing PRRs is the C-type lectin family, which mainly recognizes fungal cell wall carbohydrates[10]. An important group within this family is the NK cell lectin-like C-type lectin receptors. This group includes Dectin-1, one of the best characterized fungal PRRs, which recognizes β-glucan on fungal cell walls but is mainly expressed on myeloid immune cells and some groups of T, but not NK cells[10,12,20].

NKG2D is another lectin-like protein of the NK cell lectin-like C-type lectin receptor group. This receptor is expressed on most NK cells, cytotoxic T cells, γδ T cells, and NKT cells, and also on T helper cells under certain conditions[21]. While its structure implies a potential carbohydrate-binding ability, all of its known ligands are self-proteins. These are expressed on the cell surface upon stress, and their recognition is glycosylation independent[21–24]. Several human NKG2D ligands are known; MICA, MICB, and 6 members of the UL16 binding protein family (ULBP1-6). NKG2D ligands are expressed on mammalian cells upon sensing of stresses such as viral infection or cancerous transformation[21,22]. Recognition of these ligands by NKG2D leads to immune activation and elimination of the NKG2D-ligand-expressing cells. In addition, NKG2D is expressed during education and maturation of NK cells and can regulate the expression of additional receptors[25]. NKG2D has been implicated as a crucial receptor in the lymphocyte response to various viral and cancerous threats, both in vitro and in vivo, using NKG2D-deficient mice models[21,22,25–27].

Although many of its C-type lectin family members are major fungal PRRs, NKG2D has not been implicated in the direct recognition of fungal targets specifically, or of any other microbial or non-self targets.

In this study, we show that NKG2D is an anti-fungal PRR. NKG2D recognizes diverse fungal pathogens such as *Candida albicans*, *Cryptococcus neoformans*, and *Aspergillus fumigatus*, and this recognition leads to intracellular signaling, immune cell activation, and direct elimination of fungi. We demonstrate that NKG2D is pivotal in vivo for fungal elimination, using models of both invasive and mucosal fungal infections.

## Results

### NKG2D binds and mediates elimination of *C. albicans* cells

To identify fungal-recognizing NK receptors, we generated a library of recombinant fusion proteins composed of the ligand-binding extracellular domains of various NK receptors fused to the Fc domain of human IgG1. The proteins were incubated with yeast cells of the model fungal pathogen *C. albicans*, and binding was analyzed using flow cytometry. While our Ig-tagged negative control protein (NKp46-D1-Ig) or a 2nd tested NK receptor (CD16-Ig) did not bind *C. albicans*, NKG2D-Ig bound *C. albicans* cells specifically (Fig. 1A, quantified in B). To validate the specificity of this interaction, we also generated a murine NKG2D-Ig fusion protein and observed similar *C. albicans* binding (Fig. 1C, quantified in D).

We next examined the functionality of the *Candida*-NKG2D interaction. We isolated primary NK cells from healthy human donors and co-incubated them with *C. albicans* cells. Following co-incubation, we plated the cells on Sabouraud agar plates, counted the number of *C. albicans* colonies, and compared them to fungal cells that were grown without NK cells (schematically described in Fig. 1E). The experiment was performed in the presence of NKG2D-blocking antibodies or isotype control antibodies.

Anti-fungal cytotoxic activity was observed in the presence of isotype control antibodies but was significantly reduced in the presence of an NKG2D blocking antibody (Fig. 1F), indicating that NKG2D is involved in NK cell anti-fungal activity.

NK cells directly kill their targets by the release of granules containing various cytotoxic proteins. Granule-dependent killing was shown to be effective against various targets, including fungal ones[28]. During granule release, the membrane protein CD107a that is found in the lytic granules appears on the cell membrane. To test whether the release of cytotoxic granules is NKG2D-dependent, we repeated the co-incubation experiment, this time isolating the surviving NK cells and staining for CD107a on the plasma membrane. As expected, co-incubation of NK cells with *C. albicans* led to increased CD107a levels in the NK cells (Fig. 1G). The addition of an NKG2D-blocking antibody significantly reduced CD107a levels, indicating that NKG2D-mediated de-granulation is involved in NK cell response to *C. albicans*.

Human NK cells are classically divided into two main subpopulations according to the expression levels of the surface marker CD56: CD56dim cells and CD56bright cells. These cells are also functionally distinct. CD56dim cells are more prevalent in the blood and cytotoxic, while CD56bright cells are more common in lymphoid tissues, and secrete significant amounts of pro-inflammatory cytokines such as IFNγ[29,30]. To test which of these populations responded to *C. albicans* and to further investigate whether freshly isolated NK cells also kill *C. albicans*, we isolated NK cells from the fresh blood of several human donors and co-incubated them with *C. albicans* cells. Following co-incubation, the cells were stained for CD56 and CD107a and analyzed using flow cytometry. Both the CD56dim and CD56bright populations showed significant degranulation following *C. albicans* recognition (Fig. 1H).

We also tested whether the levels of CD107a upregulation correlate with NKG2D expression following *Candida* recognition. We stained CD56dim and CD56bright NK cells for NKG2D and CD107a following co-incubation with *C. albicans* and calculated the $R^2$ values between the two proteins (Fig. 1I, representative graphs and gating strategy presented in Supplementary Figure 1A). We found no correlation between the levels of these two proteins.

Finally, we examined whether the NKG2D-mediated fungal recognition leads to direct NKG2D activation and downstream intracellular signaling. If Candida recognition by NKG2D leads to intracellular signaling, it is most likely mediated by DAP10 as it is the only known NKG2D signaling adapter in humans. Following NKG2D activation, DAP10 is tyrosine-phosphorylated and recruits the Vav1/Grb2 complex and p85 of the PI3K complex[21,22]. To test this, we co-incubated NK cells with *C. albicans* cells for 5, 15, and 45 min, or kept them alone (0 min of incubation). The cultures were then lysed and immunoprecipitated using anti-DAP10 antibodies. The immunoprecipitate was purified, run on a gel, and stained with antibodies against either DAP10 or phospho-tyrosine (Fig. 1J). In these blots, we identified 3 main bands (10, 37 and 60 kDa). While the 10 kDa band corresponds to the previously described and predicted size of DAP10[31], the two other bands probably correspond to heavily glycosylated forms and/or immunoprecipitated complexes of DAP10 with its other signaling and regulatory partners. To assess the levels of NKG2D signaling and DAP10 phosphorylation, we also quantified the phospho-DAP10 to DAP10 ratio (Fig. 1K). As can be seen, all three band sizes were phosphorylated following activation with *C. albicans*, but the main effect was observed in the well-described 10 kDa band, which probably corresponds to free DAP10.

These results demonstrate that NKG2D binds *C. albicans* cells, which leads to NKG2D activation and downstream signaling, to NK cell activation and degranulation, and eventually to the killing of the target fungal cell.

### NKG2D mediates anti-fungal activity against a broad variety of fungal pathogens

We next tested the target range of the NKG2D-Ig *Candida* interaction. Several non-albicans *Candida* species were stained with human

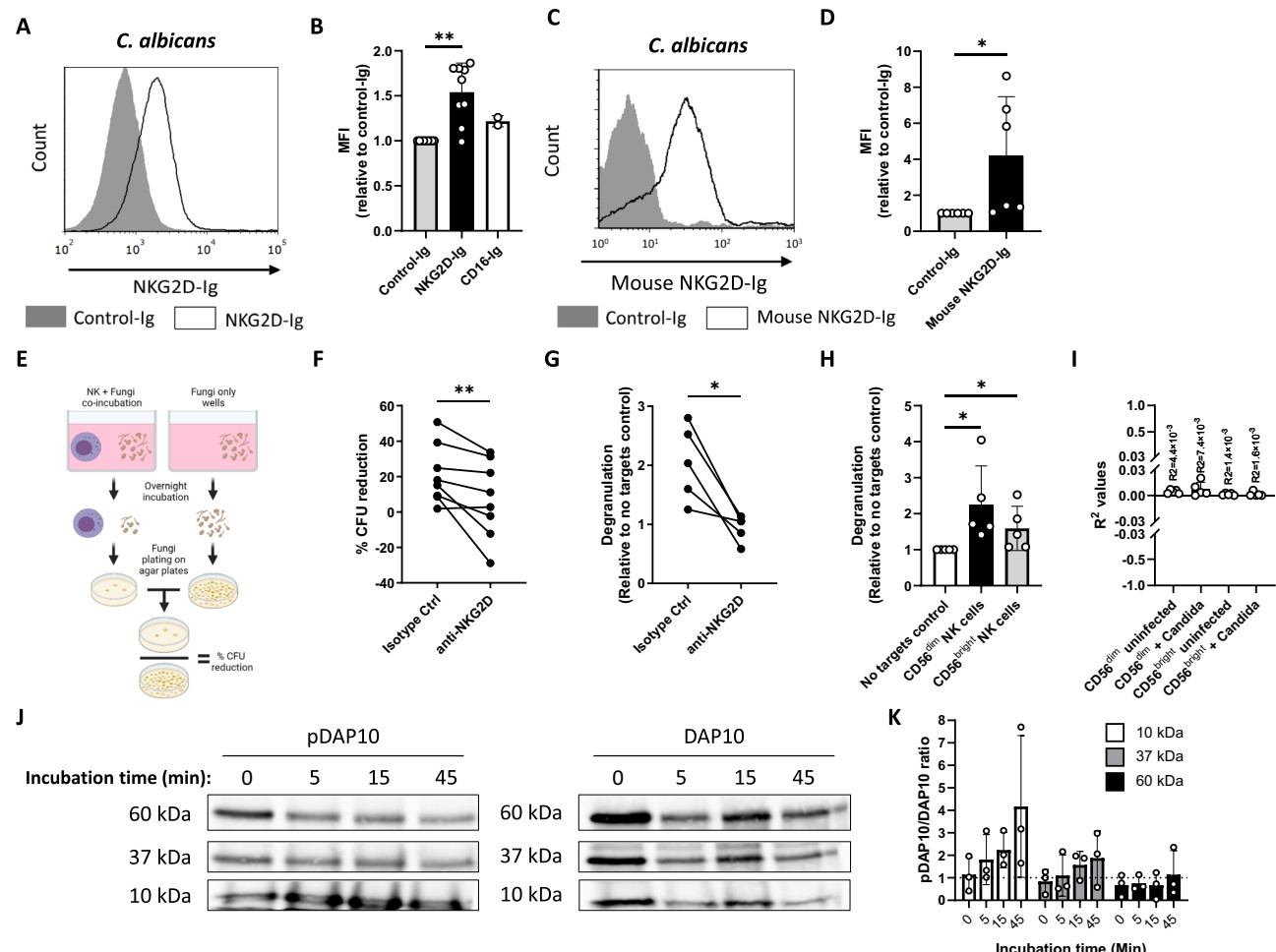

**Fig. 1 | NKG2D binds and mediates elimination of C. albicans cells. A** Flow cytometry staining of *C. albicans* yeasts using human NKG2D-Ig or a Control-Ig. One representative experiment out of 9 is presented. **B** Quantification of Ig-fusion protein staining followed by flow cytometry of *C. albicans* yeast. Presented are stainings using Control-Ig, NKG2D-Ig and CD16-Ig. $p = 0.0078$. **C** Flow cytometry staining of *C. albicans* yeasts using mouse NKG2D-Ig or a Control-Ig. One representative experiment out of 6 is presented. **D** Quantification of the results presented in C. $p = 0.03$. **E** A diagram depicting the in vitro anti-fungal NK cytotoxicity model in (**F**). **F** Anti-fungal NK cytotoxicity against *C. albicans* using NK cells blocked with α-NKG2D or isotype control antibodies. Each line represents NK cells isolated from a single independent donor. $n = 8$, $p = 0.0078$. **G** NK cell degranulation assay against *C. albicans* in the presence of α-NKG2D antibodies or an isotype control. Shown is the relative change in surface CD107a expression relative to NK cells that were not co-incubated with fungal cells. Each line represents NK cells isolated from a single independent donor. $n = 5$, $p = 0.03$. **H** NK cell degranulation assay against *C. albicans* was performed using fresh NK cells stained

and gated from the CD56[dim] and the CD56[bright] sub-populations. Shown is the relative change in surface CD107a expression relative to NK cells that were not co-incubated with fungal cells. Each dot represents NK cells isolated from a single independent donor. $n = 5$, $p = 0.03$ for both sub-populations. **I** $R^2$ values measuring the correlation between NKG2D expression and CD107 upregulation, derived from the data presented in (**H**). Each dot represents NK cells isolated from a single independent donor. **J** Immunoblots of NK cell lysates immunoprecipitated with α-DAP10 and stained with different antibodies against DAP10 or antibodies against phospho-tyrosine (pDAP10). Lysates of cells incubated with *C. albicans* cells for 0, 5, 15, or 45 min are shown. One representative experiment out of 3 is presented. **K** Quantification of results in (**J**). Data for (**B**, **D**, **H**, and **K**) are presented as mean ± SEM. Significance was tested using the Wilcoxon signed-rank test. In data for (**B**, **D**), a two-sided test was performed, and for data for (**F**, **G**, **H**) one-sided test was used. * = $p < 0.05$, ** = $p < 0.01$. Figure 1E Created in BioRender. Chaouat, A. (2024) BioRender.com/k73p450.

NKG2D-Ig and analyzed using flow cytometry. NKG2D-Ig recognized three species: *C. glabrata*, *C. parapsilosis*, and *C. tropicalis*, while a fourth one, *C. krusei*, was not recognized reproducibly (Fig. 2A–E). Thus, we concluded that NKG2D binds various *Candida* species. To test whether this binding is functional, we chose to work with *C. glabrata*. *C. glabrata* was chosen as it is the 2nd most common cause of candidiasis in many settings and is evolutionary and genetically far from *C. albicans*[32,33]. Similar to our previous experiments, we co-incubated NK cells with *C. glabrata* cells in the presence or absence of NKG2D-blocking antibodies. We then measured fungal elimination by NK cells by plating the co-cultures on Sabouraud agar plates and counting the developing fungal colonies. As can be seen (Fig. 2F), NK cells recognized and killed *C. glabrata* cells, and this

process was mediated by NKG2D. As NKG2D appears to recognize several *Candida* species, we next wondered how broad fungal recognition by NKG2D is. We used NKG2D-Ig proteins to examine NKG2D binding to another central fungal pathogen; *Cryptococcus neoformans*. We isolated and examined two *Cr. neoformans* strains from human patients and used an ELISA assay to measure NKG2D-Ig binding. Both strains were significantly stained by NKG2D-Ig (Fig. 2G, H). We then examined the functionality of this interaction by repeating the experiments described in Figs. 1F and 2F but with *Cr. neoformans* cells as targets. Incubation with primary human NK cells led to the killing of both strains, and this activity was significantly diminished in the presence of an NKG2D-blocking antibody (Fig. 2I, J).

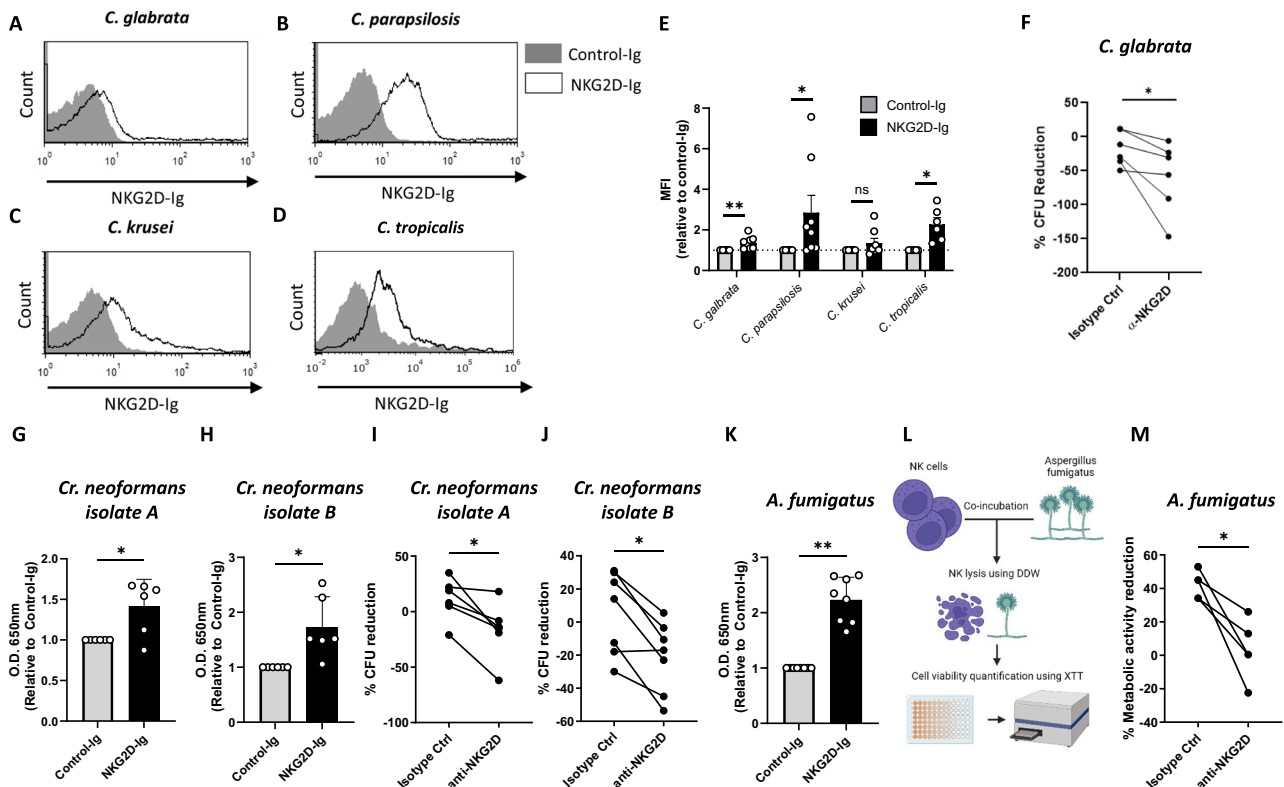

**Fig. 2 | NKG2D mediates anti-fungal activity against a broad variety of fungal pathogens. A–D** Flow cytometry staining of *C. glabrata* (**A**), *C. parapsilosis* (**B**), *C. krusei* (**C**), or *C. tropicalis* (**D**) yeasts using human NKG2D-Ig or a Control-Ig protein. One representative experiment out of 6–8 is presented. **E** Quantification of the results in (**A–D**). Each dot represents an independent experiment with fresh fungi. *n* = 6 for *C. tropicalis* (*p* = 0.03) and *n* = 8 for *C. glabrata* (*p* = 0.0078) *C. parapsilosis* (*p* = 0.0156) *C. krusei* (*p* = 0.148). Data are presented as mean values +/− SEM**. F** Anti-fungal NK cytotoxicity against *C. glabrata* using NK cells blocked with α-NKG2D or isotype control antibodies. Each line represents NK cells isolated from a single independent donor. *n* = 6, *p* = 0.0156. **G, H** ELISA assay with plate-fixed *Cr. neoformans* cells from two independent clinical isolates stained with human NKG2D-Ig or a control-Ig. Each dot represents an independent staining of freshly thawed *Cr. Neoformans*. *n* = 6, *p* = 0.0469 (**G**), *p* = 0.0156 (**H**). Data are presented as mean values +/− SD. **I, J** Anti-fungal NK cytotoxicity against two independent clinical isolates of *Cr. neoformans* using NK cells blocked using α-NKG2D or isotype control antibodies. Each line represents NK cells isolated from a single independent donor. *n* = 6, *p* = 0.0156 (**I**) and *n* = 7, *p* = 0.0156 (**J**). **K** ELISA assay with plate-fixed *A. fumigatus* conidia stained with human NKG2D-Ig or a control-Ig. Each dot represents the staining of an independent *a. fumigatus* or an individually generated NKG2D-Ig. *n* = 8, *p* = 0.0039. Data are presented as mean values +/− SD. **L** A diagram depicting the in vitro anti-fungal XTT model used in (**M**). **M** Anti-fungal activity assay using the cell viability reporter dye XTT. NK cells were incubated with *A. fumigatus* hyphae in the presence of NKG2D-blocking antibodies or isotype control, followed by their clearance using hypotonic double de-ionized water. Survival rates of the fungal hyphae were examined using XTT and are presented relative to fungal cultures that were not exposed to NK cells. Significance was tested using the Wilcoxon matched-pairs signed rank test. For data in (**E**), the two-tailed test was performed, and for (**F–K**, **M**) one-tailed test was used. ns = not significant. * = *p* < 0.05, ** = *p* < 0.01. Figure 2L was Created in BioRender. Chaouat, (A). (2024) BioRender.com/j55q074.

Another important human fungal pathogen is the mold *Aspergillus fumigatus*. As this fungus has a different lifestyle and physical properties relative to both *Candida* and *Cryptococcus*, we decided to also test whether it is recognized by NKG2D. *A. fumigatus* grows mostly as a mold containing an interconnected network of hyphae rather than as single yeast cells, so we decided to use an alternative method instead of the colony-based experiment described in Fig. 1 to check the ability of NKG2D to mediate a response against it. We first checked whether NKG2D binds *A. fumigatus* cells. To that end, we fixed *A. fumigatus* conidia and stained them with NKG2D-Ig or control-Ig proteins using an ELISA assay, similar to the process described for Fig. 2G + H. As was observed for *Candida* species and for *Cr. neoformans*, NKG2D also bound *A. fumigatus* cells significantly more than the control protein (Fig. 2K). In order to see if this interaction is functional, we seeded *A. fumigatus* conidia for 12 h and then co-incubated the grown hyphae with primary human NK cells. Following co-incubation, we eliminated the NK cells using double de-ionized water and quantified the amount of surviving fungal cells by measuring their metabolic activity using the tetrazolium salt XTT (Fig. 2L). NK cells were able to significantly reduce the metabolic activity of *A. fumigatus* cells in the presence of an isotype control antibody, but as observed with *Candida*

and *Cryptococcus*, such ability was significantly reduced when an NKG2D-blocking antibody was added (Fig. 2M).

Finally, we attempted to identify the fungal ligands or ligands of NKG2D. All known NKG2D ligands are proteins[21,22], but these are all mammalian self-ligands. However, most fungal PRRs, including members of the C-type lectin-like family (which NKG2D is also a part of), recognize carbohydrates on the fungal cell wall[10]. Since NKG2D recognizes vastly different fungi, we hypothesized that the ligand might be a common glycan and not a protein, as one highly conserved ligand is more likely than a group of ligands, and common cell-wall carbohydrates tend to be better conserved than proteins across the diverse fungal evolutionary tree. To test this, we incubated NKG2D-Ig with a microarray containing 672 lipid-linked carbohydrates from various fungal, bacterial, and plant species (full list of probes provided as Supplementary data 1). Unfortunately, no significant binding of NKG2D-Ig was detected (Supplementary Figure 2).

In summary, we found that NKG2D recognizes diverse fungal targets across the fungal family tree, including *Candida* species, *Cryptococcus neoformans*, and *Aspergillus fumigatus*, but the identity of the responsible ligand or ligands remains to be discovered.

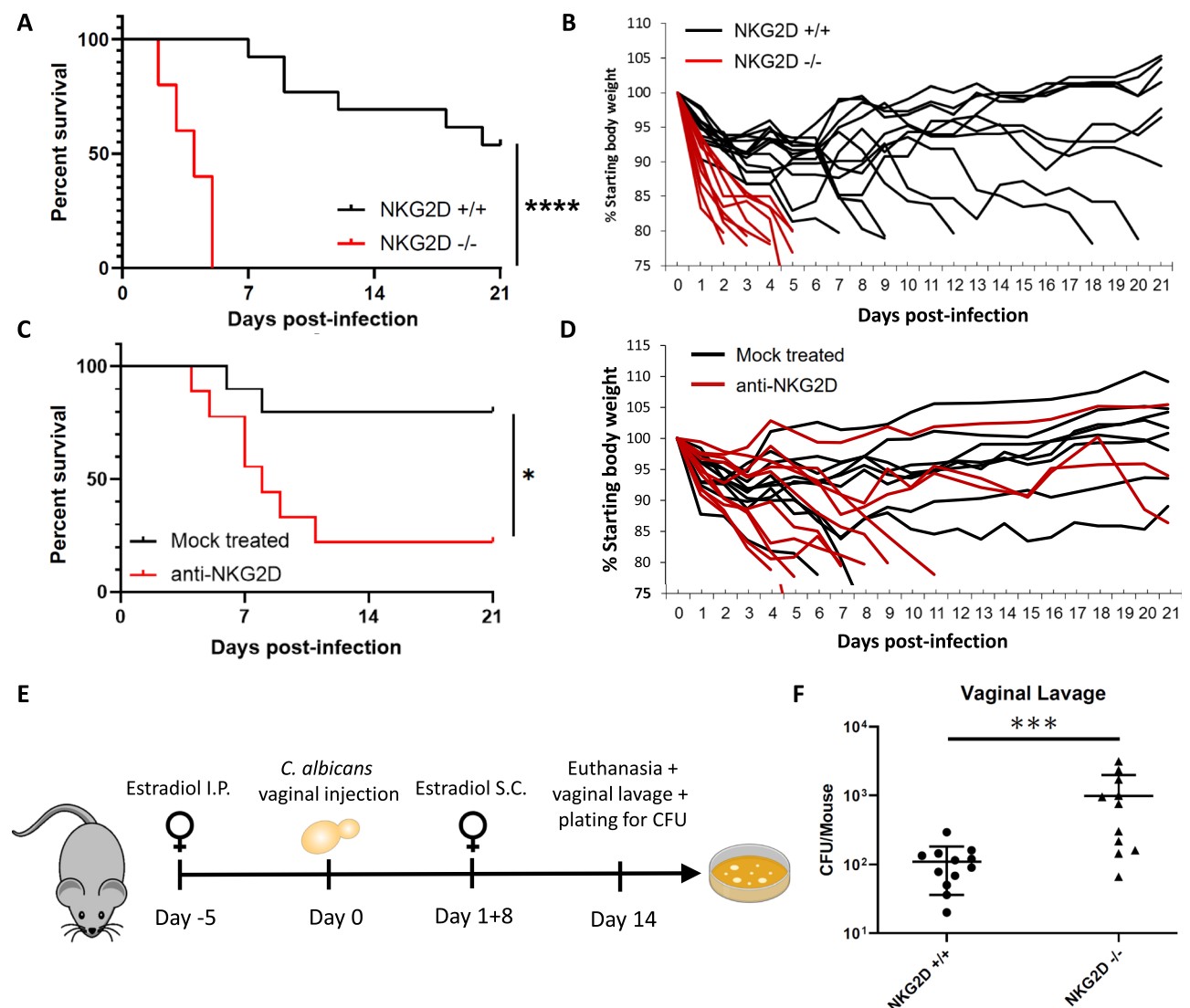

**Fig. 3 | NKG2D is critical for the in vivo control of *Candida* infections. A** Survival of C57BL/6 mice (black line) or NKG2D KO mice (red line) following intravenous injection of *C. albicans* cells. Each line represents 10–13 mice from 2 independent experiments. *p* < 0.0001. **B** Change in body weight as a measure of disease severity of C57BL/6 mice (black lines) or NKG2D KO mice (red lines) following intravenous injection of *C. albicans* cells. Each line represents a single mouse. Data is shared and derived from the same experiments as (**A**). **C** Survival of C57BL/6 mice mock-treated (black line) or treated with NKG2D blocking antibodies (red line) following intravenous injection of *C. albicans* cells. Each line represents 9 or 10 mice from 2 independent experiments. *p* = 0.0221. **D** Change in body weight as a measure of disease severity of C57BL/6 mice mock-treated (black line) or treated with NKG2D blocking antibodies (red line) following intravenous injection of *C. albicans* cells. Each line represents a single mouse. Data is shared and derived from the same experiments as (**C**). **E** A diagram depicting the in vivo candida vulvovaginitis model used in (**F**). **F** *C. albicans* burden in vaginal lavage of C57BL/6 or NKG2D KO mice 2 weeks post-inoculation. *n* = 11 (wt) or *n* = 12 (NKG2D KO) mice examined over 2 independent experiments. *p* = 0.0007. Data are presented as mean values +/− SD. For the survival assays significance was tested using the Mantel−Cox log-rank test. For fungal burden, significance was tested using the two-tailed Mann-Whitney test. * = *p* < 0.05, *** = *p* < 0.001, **** = *p* < 0.0001. Figure 3E was Created in BioRender. Chaouat, A. (2024) BioRender.com/w89b021.

## NKG2D is critical for the in vivo control of *Candida* infections

We next tested whether fungal recognition by NKG2D is relevant in vivo. To that end, we used a mouse model for disseminated candidiasis in which *C. albicans* cells are injected into mice intravenously. To that end we used a strain of NKG2D KO (*Klrk1*⁻/⁻) mice that have been previously described[26]. These mice have a normal phenotype while unchallenged and do not have an overt pathology in NK and T cell development but are more susceptible in models for various malignancies[26]. A similar mouse strain created in an independent lab also found that these mice have some changes in NK cell development, and relative resistance in a model for Murine Cytomegalovirus infection[25,27]. We infected both WT and NKG2D KO mice and followed the clinical course of their disease. Remarkably, NKG2D KO mice were extremely sensitive to *C. albicans* infection. Under conditions in which

roughly half the WT mice gradually lost weight and reached the experimental humane endpoint within 3 weeks, all the NKG2D KO mice abruptly lost weight and had to be euthanized within 5 days (Fig. 3A, B).

NKG2D is expressed on lymphocytes at various stages of development and, under certain conditions, can affect the activation threshold of other receptors such as NKp46[21,25]. Therefore, to validate the in vivo effect of NKG2D against *C. albicans* infection, we repeated the disseminated candidiasis model using WT mice that were injected with an NKG2D-blocking antibody or mock-treated. Injection of anti-NKG2D led to significantly increased susceptibility to *Candida* infection relative to the mock-treated mice (Fig. 3C, D).

*Candida* infection can lead to various diseases. While disseminated candidiasis is among the deadliest, mucosal *Candida* infections, specifically vulvovaginal candidiasis, are the most common ones. As such, we

were interested in checking whether NKG2D plays a role in mucosal *Candida* infections. To that end, we utilized a model for murine vulvo-vaginal candidiasis in which we pre-treated female mice with estradiol to increase their susceptibility to vaginal *C. albicans* infection and then inoculate them with yeasts[34]. The mice were followed, and 2 weeks post-inoculation vaginal fungal burden was quantified (Fig. 3E). Similar to our observations regarding invasive infections, NKG2D KO mice were also significantly more susceptible to vulvovaginal *Candida* infection as seen by their increased fungal burden (Fig. 3F).

In conclusion, NKG2D plays a critical role in the immune response to both invasive and mucosal *C. albicans* infections, and mice lacking NKG2D are extremely susceptible to such diseases.

### NK and T cells use NKG2D to eliminate Candida cells in vivo

The increased susceptibility of NKG2D KO mice to invasive *Candida* infections can be due to either reduced immune response resulting in diminished fungal clearance or exaggerated immune recognition leading to immunopathology. To examine whether the absence of NKG2D indeed leads to less efficient immune control of fungal infections, we repeated the in vivo invasive candidiasis model described above, this time ending the experiment on day 2 post-infection and harvesting various relevant organs. These organs were grounded and plated on Sabouraud agar plates for measurement of fungal burden. Diminished fungal clearance should lead to increased fungal burden in NKG2D KO mice, while overactivation of the immune response should not increase this value and might even reduce it. In this mouse model, the central site of infection is the kidneys[35]. Indeed, we observed the highest fungal burden at this site (Fig. 4A). Remarkably, a significant 100-fold increase in fungal burden was found in the kidneys of NKG2D KO mice (Fig. 4A), with a similar trend observed in the liver as well (Fig. 4B).

We next tested which immune cell population is responsible for the NKG2D-mediated effect. Two main immune cells are known to express NKG2D; T cells and NK cells[21]. Both cell types are also known to be involved in anti-fungal immunity and play a critical role in this specific model[19]. We repeated the in vivo fungal burden experiments, this time depleting either T or NK cells using antibodies. Surprisingly, the NKG2D KO mice were still more susceptible to infection following depletion of either T or NK cells (Fig. 4C). We therefore depleted both T and NK cells, which led to equal fungal burden and an identical phenotype in the WT and in NKG2D KO mice (Fig. 4C). These results led us to the conclusion that both T and NK cells use NKG2D in vivo to eliminate invading fungal cells.

The phenotype we observed in NKG2D KO mice can be either due to reduced NK and T cell activity against fungal cells, or due to reduced cell numbers secondary to developmental or localization errors caused by the genetic manipulation. To test this, we again infected WT and NKG2D KO mice with *C. albicans* and harvested relevant organs (kidneys, blood, and liver) on day 2 of infection. Leukocytes were isolated from the organs, stained with antibodies against NK, T helper, and cytotoxic T cell markers, and quantified using flow cytometry (Fig. 4D–F, representative graphs and gating strategy in Supplementary Fig. 1B, C). As can be seen, no significant reduction in the basal levels of these immune cell populations was identified in any of these organs. We did observe some reduction in the number of T cells (both $CD8^+$ and $CD4^+$), but not NK cells, in the liver post-infection in KO relative to WT mice. In the main site of infection, the kidneys, we observed a significant increase in the basal levels of all three lympho-cyte populations in KO mice. During infection, this increase was still maintained for the NK cells, but not for the T cells. Although we observed interesting changes in lymphocyte immune populations during infection of NKG2D KO mice with *C. albicans*, the absence of NKG2D-expressing cells (mainly NK and cytotoxic T cells) was not noticed. This supported our hypothesis that the NKG2D KO mice sensitivity to fungal infections is probably due to immune cell dys-function and not to reduced cell numbers.

Finally, we asked whether this anti-fungal activity is mediated by direct interaction between these cells and fungal targets, as we observed in vitro for human cells using NKG2D-blocking antibodies (Fig. 1 + 2). To that end, we isolated NK cells from spleens of WT and NKG2D KO mice and co-incubated them with *C. albicans* cells. NK cells from NKG2D KO mice were significantly impaired in their ability to eliminate fungal cells ex-vivo (Fig. 4G).

In conclusion, these results point to a significant role of NKG2D, expressed in NK and T cells, in the recognition and direct elimination of fungal pathogens.

## Discussion

This study provides evidence that the activating receptor NKG2D is a fungal PRR able to recognize and eliminate a diverse range of fungal pathogens, including clinically relevant *Candida*, *Cryptococcus* and *Aspergillus* species. The effects of NKG2D were observed both in vitro and in vivo, and in models for both invasive and mucosal infections.

NKG2D is known to be an important receptor, but so far only in NK and T cell responses to viral and malignant threats. During various stresses, NKG2D ligands are expressed on the cell surface of self-cells marked for NK- or T-cell mediated elimination[21,22]. Our data suggests that NKG2D is also a powerful anti-fungal PRR, recognizing diverse fungal species. Although our findings shed light on lymphocyte-mediated anti-fungal NKG2D-mediated immune response, many questions remain unanswered; The properties of the lymphocyte-fungal immune synapse, the identity of the specific cytotoxic proteins responsible for this effect, or the intracellular mechanism responsible for the death of the target fungal cell.

This is especially emphasized in the case of $CD8^+$ T cells, as their classic recognition mechanism of mammalian targets requires MHC restriction, and such MHC molecules and antigen presentation are not known to be present in fungal cells. While we have shown that T cells play an important NKG2D-mediated role in our murine model of *C. albicans* infection[19] (Fig. 4C), their mode of activation and target recognition remains unknown. A possible mechanism is the involve-ment of TCR-independent activation of bystander $CD8^+$ T cells. This phenomenon has been observed in various viral and bacterial infec-tions and in cancer models, but not in the context of fungal infections. Following induction by pro-inflammatory cytokines such as IL-12, IL-15, and IL-18, these "bystander" T cells recognize targets using NK recep-tors, mainly through NKG2D[36,37]. As the cytokines required for the induction of these cells are present and are known to have a role in anti-Candida imunity[33,38], and as they use NKG2D as one of their main receptors, the involvement of bystander CD8 + T cells seem to be a promising hypothesis.

Since NKG2D is a c-type lectin-like receptor, and many members of this receptor family, such as Dectin-1 or MelLec, are central fungal PRRs[20,39], it is possible that fungal recognition is actually the original role of NKG2D, and its well-described stress-recognition functions only emerged later. Several other closely related members of the NKG2/CD94 family currently exist as orphan receptors without clear func-tions or ligands[40]. Given their great similarity to their well-known relative NKG2D, many studies so far focused on identifying their cel-lular targets. In light of the anti-fungal role of NKG2D we describe here, and our hypothesis that fungal recognition could be an ancient NKG2D function, it is possible that the functions of other NKG2 proteins include fungal recognition.

Our work demonstrates that NKG2D can directly recognize fungal cells, initiate immune activation and direct fungal killing in both human and murine lymphocytes. While this was shown both in vitro and ex-vivo, it is possible that NKG2D fulfills additional roles during in vivo fungal infections, for example, by activation of lymphocytes by self-ligands on mammalian cells during infection. While possible, such an additional role is hard to study without first identifying the identity of the fungal ligand and removing it from the in vivo models.

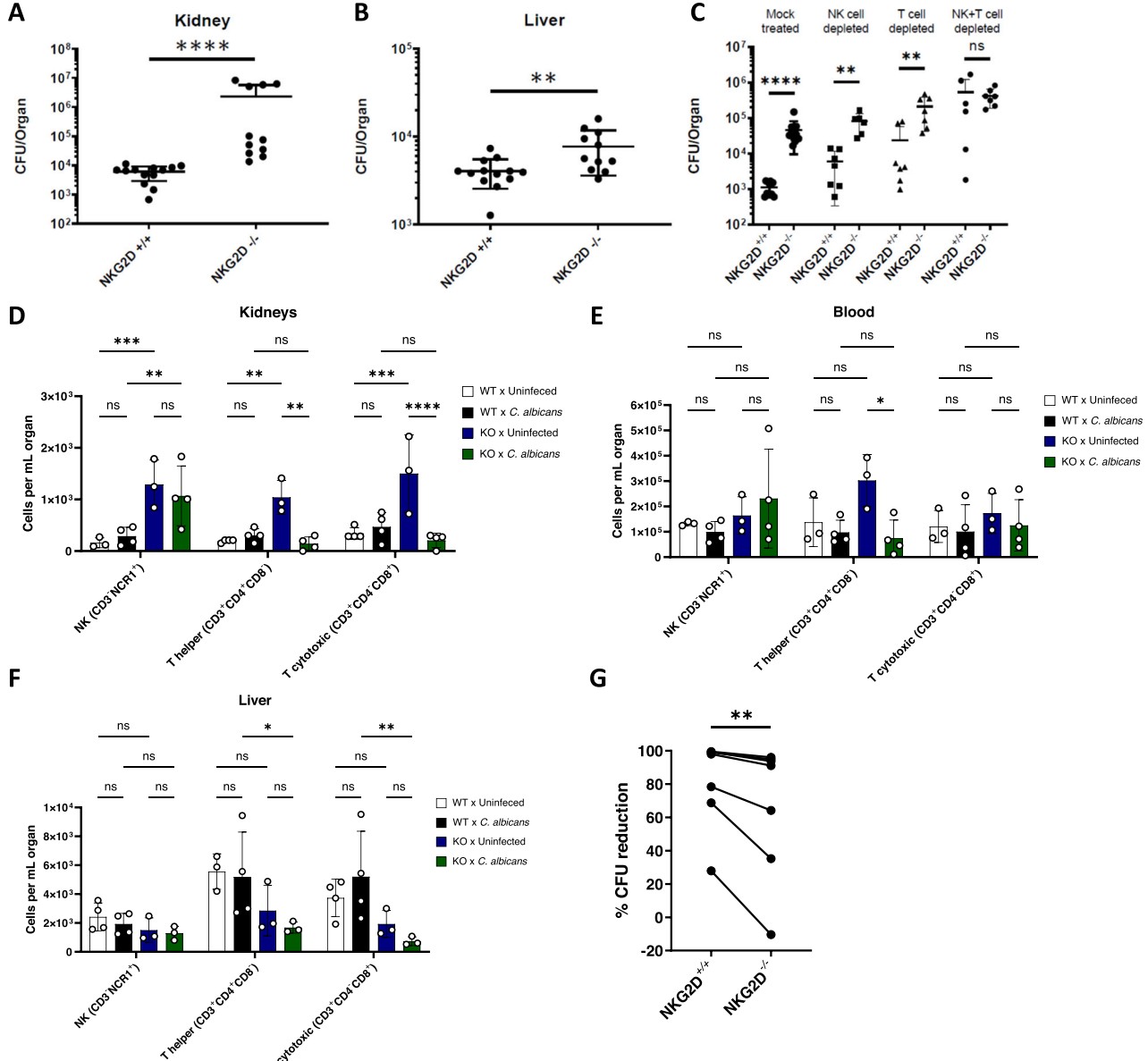

**Fig. 4 | NK and T cells use NKG2D to eliminate Candida cells in vivo. A, B.** Fungal burden in the kidneys (**A**) or livers (**B**) of C57BL/6 or NKG2D KO mice 48 hours after I.V. infection with *C. albicans* yeasts. The organs were harvested, processed, and seeded on Sabouraud agar plates. $n = 11$ (NKG2D KO) or $n = 13$ (wt) mice examined over 2 independent experiments. $p < 0.0001$(A), $p = 0.0076$ (**B**). Data are presented as mean values +/− SD. **C** Fungal burden in the kidneys of C57BL/6 or NKG2D KO mice 48 h after I.V. infection with *C. albicans* yeasts. The mice were injected intraperitoneally with depleting antibodies against NK cells, T cells, or both on days − 1 and 1 post-infection. The organs were harvested, processed, and seeded on Sabouraud agar plates. $n = 10$ (Mock treated wt), $n = 11$ (Mock treated NKG2D KO), $n = 7$ (NK depleted wt and T cell-depleted wt and NKG2D KO, NK and T cell Depleted NKG2D KO), $n = 6$ (NK depleted NKG2D KO, NK and T cell Depleted wt), group examined over 2

independent experiments. Data are presented as mean values +/− SD. **D–F.** Leukocytes were isolated from kidneys (**D**), blood (**E**), or livers (**F**) of uninfected or *C. albicans*-infected NKG2D WT or KO mice. The cells were stained with antibodies against CD3 and NCR1 or CD3, CD4, and CD8 and divided into NK (CD3⁻NCR1⁺), T helper (CD3⁺CD4⁺CD8⁻), and T cytotoxic cells (CD3⁺CD4⁻CD8⁺). Data is presented as mean ± SEM. **G** Anti-fungal NK cytotoxicity against *C. albicans* using splenic NK cells isolated from WT or NKG2D KO mice. Each line represents an average of at least 3 repeats from NK cells isolated from a single mouse. $p = 0.0078$. Significance was tested using the two-tailed Mann-Whitney test (**A–C**), two-way ANOVA with Tukey's multiple comparisons test (**D–F**), or one-tailed Wilcoxon matched-pairs signed rank test (**G**). ns = not significant, * = $p < 0.05$, ** = $p < 0.01$, *** = $p < 0.001$, **** = $p < 0.0001$.

Whether by recognition of the fungal ligand or classical self-ligands, it is possible that this interaction plays additional roles in the anti-fungal immune response. Such possible effects could be control of exaggerated immune responses leading to immunopathology or sculpting of the fungal microbiome during health and disease. While certainly possible, especially given the extreme phenotype we observed in our in vivo experiments, we did not examine such possibilities, and they remain an interesting subject for future studies.

A central question regarding the role of NKG2D as a fungal PRR is the identity of its ligand or ligands. All of the NKG2D ligands described so far are self proteins[21,22]. As such, it is likely that a protein, probably one located on the fungal cell wall, is the NKG2D fungal ligand. Nevertheless, most other closely related c-type lectin-like fungal PRRs recognize carbohydrate moieties such as β-glucan or Mannose[10]. The varied pool of fungi recognized by NKG2D seems to hint at a highly conserved element as the ligand. Therefore, it is possible that NKG2D

recognizes fungal carbohydrates, as they are usually more structurally conserved between species when compared to proteins. While we were not successful in identifying a glycan ligand, it is possible that it was not one of the specific probes we analyzed, or that an additional factor missing from the experiment is needed for this interaction. Additional possibilities for the ligand identity also exist, such as an unrelated molecule, as in the case of Melanin recognition by the c-type lectin MelLec[39]. As many pathogens developed mechanisms to evade immunity, it is very likely that fungal immune evasion mechanisms exist to evade NKG2D-mediated recognition. Interestingly, a previous study found that *A. fumigatus* is able to downregulate NKG2D on NK cells following the generation of an immune synapse[41]. Previous studies showed that NK recognition of *A. fumigatus* is morphology-dependent and that NK cells are mostly potent against hyphae, and not conidia[42]. Our findings demonstrate NKG2D binding using ELISA (Fig. 2K), in which the fungal population contains mainly conidia, and NKG2D activity using the XTT metabolic activity assay (Fig. 2M), in which the fungal population is mostly composed of hyphae. While both populations contain some presence of both morphologies, it hints that NKG2D can recognize both morphologies and is mediating cytotoxicity, at least against hyphae, and possibly to some extent against conidia as well. Another potential immune evasion mechanism might be employed by *C. krusei* cells, as we observed high variability in the ability of NKG2D-Ig proteins to bind them.

Mice lacking NKG2D were highly susceptible to fungal infections. As immune-based therapies such as CAR-T cells and antibodies increasingly enter the field of infectious diseases in general and fungal infections specifically[43], we hope that these findings could provide new therapeutic avenues against multi-drug-resistant fungal pathogens. As more and more drug-resistant fungi emerge, the patient population at risk increases, and the yearly burden of fungal-related morbidity and mortality climbs, similar developments are surely needed.

## Methods

### Ethics statement

Blood samples were collected from donors who provided informed consent in accordance with the ethical guidelines and regulations set forth by the guidelines of the institutional Helsinki committee (Helsinki number 0030-12-HMO). Mice experiments were performed in accordance with the guidelines of the Declaration of Helsinki and the local research ethics committee, as described in protocol number MD-21-16451-5.

### Primary cells and cell lines

Primary human NK cells were isolated from blood donations collected from healthy individuals. The procedure was approved by and performed according to the guidelines of the institutional Helsinki committee (Helsinki number 0030-12-HMO). The blood was treated with heparin, mixed with Lymphoprep (STEMCELL Technologies), and centrifuged to isolate peripheral blood mononuclear cells (PBMC). The EasySep human NK cell enrichment kit (STEMCELL Technologies) was used to extract NK cells from the PBMC. The cells were co-cultured in 96 well U-bottomed plates in the presence of irradiated RPMI-8866 cells ($5 \times 10^3$/well) and irradiated PBMCs from two independent donors ($5 \times 10^4$/well per donor). Cells were irradiated with 6000 RAD. The NK co-culture was grown in a 70%:30% DMEM:F12 media mix with 10% human serum (Sigma-Aldrich), 1 mM sodium pyruvate (Biological Industries), 2 mM glutamine (Biological Industries), nonessential amino acids (Biological Industries), 100 µl/ml penicillin (Biological Industries), 0.1 mg/ml streptomycin (Biological Industries), 500 µl/ml rhIL-2 (PeproTech) and 20 mg/ml PHA (Sigma-Aldrich). The cells were grown in 37 °C and 5% $CO_2$. NK cells were stained with FITC-anti-CD3 (HIT3a) and APC-anti-CD56(HCD56) antibodies (both form BioLegend) and analyzed using flow cytometry to validate their identity. Fresh NK cells were isolated and validated as

mentioned above, but instead of being co-cultured with feeder cells, they were cultured in freshly prepared DMEM:F12 media mix for 24 h and then analyzed.

Murine splenic NK cells were prepared by harvesting spleens from C57BL/6 WT or NKG2D KO (*Klrk1*$^{-/-}$) mice. The spleens were manually dissociated, and the cells were cleaned by passage through a 70 µm strainer followed by a 2-minute incubation in ACK medium (37 mM NH4Cl, 10 mM KHCO3, 0.1 M EDTA) at room temperature. EasySep mouse NK cell enrichment kit (STEMCELL Technologies) was used to isolate NK cells from splenocytes, and purity was validated using flow cytometry.

Cell lines used in this study were RPMI-8866 cells, HEK293T cells, and Vero cells (parental and MICA-expressing). RPMI-8866 cells were grown in RPMI medium (Sigma-Aldrich), and HEK293T, Vero, and Vero-MICA cells were grown in Dulbecco's modified Eagle's medium (DMEM, Sigma-Aldrich). Both culture media were supplemented with 10% inactivated fetal bovine serum (Sigma-Aldrich), 1 mM sodium pyruvate (Biological Industries), 2 mM glutamine (Biological Industries), nonessential amino acids (Biological Industries), 100 µl/ml penicillin (Biological Industries) and 0.1 mg/ml streptomycin (Biological Industries). Vero-MICA cells were generated as follows: The MICA gene was amplified from Hela cDNA HeLa using the following primers: 5'-AAAACTCGAGG<u>GCCGCCACC</u>ATGGGGCTGGGCCCGGTC-3' and 5'-TTTGGATCCTTACAACAACGGACATAGAAAATA- 3'. The PCR fragment was cloned into pHAGE-DsRED (−)-eGFP(+) lentiviral vector(Twist Bioscience) using XhoI and BamHI restriction sites. The construct was validated by DNA sequencing. Lentiviruses were generated in 293 T cells using polyethylenimine (PEI) and a transient three plasmid transfection protocol: the above-mentioned pHAGE-DsRED vector with the insert, psPAX2 (a gift from Didier Trono (Addgene plasmid # 12260)), and pCMV-VSV-G (a gift from Bob Weinberg (Addgene plasmid # 8454)).

Vero cells were infected with the lentiviral vectors, and the transduced cell population was sorted for eGFP$^+$ cells using Sony SH800S. MICA expression was evaluated using flow cytometry using an APC-anti-MICA antibody (clone #159207, R&D Systems) as described below.

### Microbial strains

The WT Candida species and strains used in this study were *Candida albicans* SC5314, *Candida glabrata* BG2, *Candida parapsilosis*, *Candida tropicalis*, and *Candida krusei*. The *Aspergillus fumigatus* strain used was a ku80 null strain on a CEA10 background. The *Cryptococcus neoformans* strains used were clinically isolated and collected by the authors.

Fungal strains were kept in glycerol stocks at − 80 °C. When needed, the strains were plated directly onto Sabouraud dextrose agar plates (30 gr/L Sabouraud Dextrose Broth (Signa-Aldrich) and 15 gr/L agar-agar). The plates were kept at 4 °C and changed regularly at intervals of 2-4 weeks. Prior to the experiment, the fungi were inoculated into sterile Sabouraud dextrose broth and grown in a shaking incubator overnight at 30 °C under aerobic conditions. *Cryptococcus* strains were then used in the relevant experiment. *Candida* overnight cultures were additionally diluted 1:50 into fresh Sabouraud dextrose broth and grown in similar conditions for an additional 3-4 h before introduction into a relevant experiment.

*Aspergillus fumigatus* frozen stocks were thawed into YAG plates. A spore stock was prepared by harvesting the plate using a 0.02% tween 20 solution and diluting the harvested fungal cells with ultra-pure water (Biological Industries). The stock was kept at 4 °C until the day of the experiment, when the cell concentration was counted using a hemocytometer and a relevant number of cells was taken for the experiment.

### Mice

The study used experiment-naïve mice aged 6-10 weeks. All experiments except the vulvovaginal candidiasis model used male mice only.

The mice were allocated randomly to the different experiments and experimental groups. All experiments were performed under specific pathogen-free (SPF) conditions in the animal facility of the Hebrew University-Hadassah Medical School (Ein-Kerem, Jerusalem) in accordance with the guidelines of the Declaration of Helsinki and the local research ethics committee.

Mice strains used were C57BL/6 and NKG2D KO (*Klrk1*[−/−]). C57BL/6 mice were purchased from Envigo. NKG2D KO mice were self-grown. The generation of the NKG2D KO mice was described previously[26].

## Flow cytometry
Cells were centrifuged and washed 3 times in ice-cold 1xPBS, 3000 G (fungi) / 515 G (mammalian cells), 4 °C for 5 min. The washed cells were counted using a hemocytometer and placed in a U-shaped 96-well plate, at a density of $1 \times 10^5$ cells/well. We next added Ig-fusion proteins (5 μg/well) or primary antibodies (0.25 μg/well) to the wells and completed the volume to 100 μl/well using FACS medium (1x PBS, 0.05% Bovine Serum Albumin, 0.05% NaN₃) and incubated the cells for 1 h on ice. The control-Ig protein used for the experiments was NKp46-Ig or its inactive derivative D1-Ig. The primary antibodies used for these experiments were APC-anti-human-CD107a (H4A3), APC-anti-human CD56 (HCD56), PE-anti-human NKG2D (1D11), BV421-anti-mouse CD3 (17A2), APC-anti-mouse NCR1 (29A1.4), APC-anti-mouse CD4 (GK1.5), APC-anti-mouse CD8 (53-6.7) and FITC-anti-mouse CD8 (53-6.7), all from BioLegend. Next, the cells were centrifuged and washed twice in ice-cold FACS medium (3000 G (fungi) / 515 G (mammalian cells), 4 °C, 5 min) and stained with a secondary antibody)0.75 μg/well) for 30 min on ice. The centrifugation-washing step was repeated twice more and followed with a re-constitution of the cells in 100 μl of FACS medium and analysis using either a FACSCalibur machine (BD Biosciences) or a CytoFlex machine (Beckman-Coulter Life Sciences) and the FCS Express software (De Novo Software).

## Fc fusion protein generation
The generation of the human and murine NKG2D-Ig and control-Ig (human NKp46-Ig or D1-Ig) fusion protein was thoroughly described previously[44–46]. Briefly, HEK293T cells exogenously expressing the fusion proteins were grown in DMEM medium as described above. Upon reaching sufficient density, the cells were transferred to a low-protein BSA-free medium (LPM, Biological Industries) complemented with 1 mM sodium pyruvate (Biological Industries), 2 mM glutamine (Biological Industries), nonessential amino acids (Biological Industries), 100 μ/ml penicillin (Biological Industries) and 0.1 mg/ml streptomycin (Biological Industries). After 3–7 days of growth and secretion of the Ig-fusion protein into the LPM medium, the medium was collected and centrifuged ($515 \times g$, 4 °C, 5 min) to clean it from surviving cells. The Ig-fusion proteins were purified from this medium using a HiTrap Protein G HP column (Sigma-Aldrich) in a BioCAD High-Pressure Perfusion Chromatography Station (PerSeptive Biosystems). Finally, the purified proteins were put in dialysis bags and buffer-exchanged to 1xPBS medium, where they were kept frozen at − 20 °C until needed experimentally. The purity of the proteins was examined using sodium dodecyl sulfate–polyacrylamide gel electrophoresis (SDS/PAGE) followed by Coomassie staining using Imperial™ Protein Stain (ThermoFisher Scientific) under both denaturing and non-denaturing conditions (Supplementary Fig. 3A). The specificity of NKG2D-Ig was also verified by staining of the low-NKG2D-ligands-expressing cell line Vero, or a daughter cell line of it in which we exogenously expressed the NKG2D ligand MICA (Supplementary Fig. 3B). In addition, NKG2D specificity was validated by plating NKG2D-Ig or control proteins on ELISA plates followed by an ELISA assay using specific anti-NKG2D (or anti-control) antibodies (Supplementary Fig. 3C). Pure proteins were quantified using Pierce™ BCA Protein Assay Kit (ThermoFisher Scientific) prior to use.

## Cytotoxicity assay
Effector (NK) and target (fungal) cells were grown as described above. The cells were centrifuged and washed 3 times in ice-cold 1X PBS (3000 G (fungi) / 515 G (mammalian cells), 4 °C, 5 min), counted using a hemocytometer, and reconstituted in RPMI medium (described above). NK cells were incubated with isotype antibody (MOPC-21, BioLegend) or anti-human NKG2D antibody (1D11, BioLegend) for 1 h on ice. One μg of antibody was used per $1 \times 10^5$ cells. Next, the effector and the target cells were mixed and co-incubated in a 96-well U-shaped plate. Each well contained $1 \times 10^3$ fungal cells and either 0 (no-effector control), $1 \times 10^5$ mammalian cells (for *C. albicans* experiments), or $0.5 \times 10^5$ mammalian cells (for *C. glabrata*), in a final volume of 200 μl RPMI medium. The cells were co-incubated for 12–14 h in a stationary 37 °C 5% $CO_2$ incubator. Following that, the cells were serially diluted in ice-cold sterile 1x PBS and plated on Sabouraud agar plates. The plates were grown in a stationary 30 °C incubator with room air for 24–48 h, and then the grown colonies were counted. Percentage CFU reduction was calculated by comparing the effector + target wells to identical wells with no effector cells.

## Immunoprecipitation
NK cells and fungal cells were grown as described above. $3 \times 10^6$ NK cells were taken per time point. Cells were washed with 1xPBS and resuspended in RPMI media (described above) and then divided into 30 wells containing $1 \times 10^5$ cells each. Fungal cells were grown and washed as described above, and co-cultured with NK cells with E:T ratio of 1:1. Co-cultured cells were collected in one 15 ml falcon tube and lysed by RIPA buffer (50 mM Tris-HCl pH 8, 150 mM NaCl, 1% NP-40, 0.5% sodium deoxycholate, 0.1% SDS) containing protease inhibitor cocktail (Sigma-Aldrich) and PMSF (Sigma-Aldrich) with 1 μM Sodium-Orthovanadate pH 10(Sigma-Aldrich). Lysates were centrifuged at $13000 \times g$, 15 min, and 4 °C to separate and remove cell debris. Supernatants were transferred to a fresh 1.5 ml microcentrifuge tube. Next, 0.4 μg anti-DAP10 antibody (H-2, Santa Cruz) was added to lysates and incubated for 1 h in 4 °C. 20 μl of resuspended volume of Protein A/G PLUS-Agarose (sc-2003, Santa Cruz) was added and incubated at 4 °C on a rotating device overnight. Immunoprecipitants were collected by centrifugation at $1000 \times g$ for 5 min at 4 °C, followed by 4 washes with 300 μl RIPA and centrifugation. After the final wash, the pellet was resuspended in 40 μl of 1x electrophoresis sample buffer (40% glycerol, 240 mM Tris/HCl pH 6.8, 8% SDS, 0.04% bromophenol blue, 5% β-mercaptoethanol) and analyzed by SDS-PAGE western blot.

## SDS-PAGE Western Blot
Protein samples in 40 μl of 1x electrophoresis sample buffer were boiled for 5 min at 95 °C. The samples were loaded onto 10% SDS-PAGE gels and separated by electrophoresis at 120 V for 2 h. Proteins were transferred onto nitrocellulose membranes using electrophoretic transfer at 300 mA for 2 h. Membranes were blocked in 5% BSA in Tris-buffered saline with 0.1% Tween-20 (TBST) for 1 h at room temperature. After blocking, membranes were incubated overnight at 4 °C with primary antibodies diluted in 5% BSA in TBST. The following primary antibodies were used: anti-phospho-Tyrosine (4G10, Cell Signaling Technology) and anti-DAP10 (H-2, Santa Cruz). The membranes were washed 3 times for 5 min each with TBST and then incubated with Peroxidase AffiniPure™ Goat Anti-Mouse IgG (H + L) secondary antibody (1:10000, Jackson ImmunoResearch) in 5% BSA in TBST for 1 h at room temperature. After washing the membranes 3 times for 5 min each with TBST, the protein bands were visualized using an enhanced chemiluminescence (ECL) detection reagent (Biological Industries) and imaged using a ChemiDoc MP Imaging System (Bio-Rad). The intensity of the protein bands was quantified using ImageJ software, and the relative protein phosphorylation levels were normalized to DAP10.

## ELISA

Fungal cells were grown as described, diluted in fresh Sabouraud Dextrose broth or Yeast Nitrogen media (for *A. fumigatus*), and then incubated for 2 h in a stationary 30 °C incubator in a high-binding clear F-bottomed 96-well ELISA plates (De-Groot group). $2 \times 10^6$ cells were plated in each well (or $5 \times 10^5$ conidia for *A. fumigatus*). Next, the cells were fixed by replacing the medium with a 1% Paraformaldehyde in 1xPBS solution and incubating the cells in it for 10–20 min. Blocking was then performed by incubating the cells in PBS-BSA (1% w/v bovine serum albumin diluted in 1xPBS) overnight at 4 °C. The cells were then stained with the Ig-fusion proteins diluted in PBS-BSA (5 μg/well) and incubated for 2 h on ice. The Ig-fusion proteins used as a negative control for the experiments were human NKp46-Ig and TIGIT-Ig. The cells were washed 4 times with PBST (1xPBS supplemented with 0.05% Tween 20). A detection antibody (Biotin-SP- AffiniPure Rabbit Anti-Human IgG, Jackson ImmunoResearch) was diluted in PBS-BSA (1:7500) and incubated with the cells for 1 h at room temperature. The plates were again washed 4 times with PBST, followed by incubation with Streptavidin-HRP (Jackson ImmunoResearch) for 30 min at room temperature and a final step of 4 washes with PBST. Finally, 3,3′,5,5′-tetramethylbenzidine (TMB) substrate (SouthernBiotech) was used to develop the plates. The resulting color reaction was read at 650 nm using a plate reader (Tecan).

For verification of Fc-fusion proteins, 5 μg/well of fusion proteins were incubated in high-binding clear F-bottomed 96-well ELISA plates (De-Groot group) overnight at 4 °C. Blocking and washes were done as described above. For the staining, 0.2 μg/well of antibodies against NKG2D (1D11), NKp44 (p44-8), and NKp46 (9E2) (all antibodies from BioLegend were added and kept for 1 h at room temperature. The following experimental steps were performed as described above.

## XTT assay

Effector (NK) and target (fungal) cells were grown as described above. The NK cells were centrifuged and washed 3 times in ice-cold 1X PBS (515 G, 4 °C, 5 min), counted using a hemocytometer, and $1 \times 10^5$ cells were incubated with 1 μg of isotype or anti-human NKG2D antibody (1D11, BioLegend) for 1 h on ice. *A. fumigatus* conidia were grown overnight in a 96-well plate with YNB medium (6.8 gr/L Yeast Nitrogen Base (Sigma-Aldrich) and 5 gr/L D-Glucose (Sigma-Aldrich)), $1 \times 10^5$ conidia/well, in a 37 °C stationary incubator with room air. The fungal plate was centrifuged ($3000 \times g$, 24 °C, 5 min), and the YNB medium was removed, followed by the addition of $1 \times 10^5$ NK cells/well in a final volume of 200 μl of RPMI + antibodies.

The effector and target cells were co-incubated for 6 hours (37 °C stationary incubator with 5% $CO_2$). Next, the mammalian cells were inactivated by 3 centrifugation/wash cycles with DDW ($3000 \times g$, 24 °C, 5 min). XTT/menadione mix was freshly prepared (1x PBS with 0.5gr/L XTT (sodium 3′-[1-[(phenylamino)-carbony]−3,4-tetrazolium]-bis(4-methoxy-6-nitro)benzene-sulfonic acid hydrate, Sigma-Aldrich) and 1 μM Menadione (Sigma-Aldrich)) and 100μl XTT was added per well. The plate was wrapped in aluminum foil and incubated for up to 2 h at 37 °C and then read in a plate reader (Tecan) at a wavelength of 490 nm.

## CD107a degranulation assay

Human NK cells were isolated from peripheral blood mononuclear cells (PBMCs) as described under"Primary cells and cell lines" section. For the CD107a assays with blocking of NKG2D, $1 \times 10^5$ cells per condition were harvested, washed 3 times with 1xPBS, and resuspended in freshly prepared RPMI-1640 media. 1 μg α-NKG2D (1D11, BioLegend) or isotype control (MOPC-21, BioLegend) per $1 \times 10^5$ cells were added, and cells were incubated on ice for 1 h.

*Candida albicans* cells were cultured overnight in Sabouraud Dextrose medium at 30 °C with shaking. The fungal cells were collected by centrifugation ($3000 \times g$, 24 °C, 5 min), washed twice with ice-cold 1xPBS, and $1 \times 10^5$ cells per well were resuspended in RPMI-1640. NK cells were co-cultured with *Candida albicans* at an effector-to-target (E:T) ratio of 1:1 in 96-well round-bottom plates. To measure degranulation, 0.2 μg of antibody per well of APC-anti-CD107a (clone H4A3, BioLegend) and FITC-anti-CD56 antibody (HCD56, BioLegend) were added directly to the wells at the beginning of the co-culture. As isotype controls, APC or FITC-conjugated MOPC-21 antibodies (Bio-Legend) were used. The cells were incubated for 5 h at 37 °C in a humidified atmosphere with 5% $CO_2$. Next, cells were centrifuged and washed twice with FACS buffer (1x PBS, 0.05% Bovine Serum Albumin, 0.05% NaN₃), resuspended in FACS buffer, and analyzed using either a FACSCalibur machine (BD Biosciences) or a CytoFlex machine (Beckman-Coulter Life Sciences) and the FCS Express software (De Novo Software).

For CD56$^{dim/bright}$ experiments, fresh NK cells were isolated as described above and cultured in freshly prepared DMEM:F12 media mix for a maximum of 24 hours prior to the experiment. $0.5 \times 10^5$ cells were taken for each condition, and the CD107a assay was performed as described above, with minor changes: $0.5 \times 10^5$ cells of *c. albicans* were co-incubated with NK cells to maintain E:T ration 1:1. PE-NKG2D (1D11, BioLegend) was used for staining in order to study the correlation between NKG2D-CD107a. PE-Isotype control(MOPC-21, BioLegend) was used as isotype control.

## Glycan microarray analysis

The glycan microarray scan for binding ligands of NKG2D was performed in collaboration with the Glycosciences Laboratory, Imperial College London, as described previously[47]. NKG2D-Ig was analyzed using Array Sets 42–56 containing 672 lipid-linked oligosaccharide probes from fungal, bacterial, and plant sources. The full probe list is provided as supplemental data 1.

In brief, the analyses were performed at room temperature. Microarray slides were blocked for 1 h using the blocker/diluent solution below: 10 mM HEPES (pH 7.4), 150 mM NaCl (referred to as HBS) containing 1% (w/v) bovine serum albumin (Sigma 7030), 0.02% Casein blocker (Pierce) and 5 mM CaCl2. The hFc-tagged NKG2D was pre-complexed with biotinylated anti-human IgG (VECTOR, BA-3000) at a ratio of 1:1 (by weight) and overlaid onto the array. The sample was tested at two overlay concentrations: 50 μg/mL followed by a 150 μg/mL concentration after no positive signal was detected using the initial concentration. The final detection was with 30 min overlay of streptavidin-Alexa Fluor 647 (Molecular Probes) at 1 μg/ml. The microarray slides were scanned with GenePix 4300 A scanner instrument (100% laser power at PMT 350) and the image analysis (quantitation) was performed with GenePix® Pro 7 software. A dedicated in-house microarray software was used for data processing and presentation.

## Murine invasive candidiasis model

Male mice and *C. albicans* cells were grown as described above. The fungal cells were washed and centrifuged 3 times in sterile ice-cold 1x PBS ($3000 \times g$, 24 °C, 5 min), counted using a hemocytometer, and diluted to a concentration of $5 \times 10^5$/100 μl in sterile 1x PBS. The mice were inoculated with the yeasts by tail vein injection, at a MOI of $5 \times 10^5$/100 μl/mouse. Following infection, mice were weighed daily. For the survival experiments a reduction of > 20% of their initial body weight was considered the ethical endpoint of the experiment and marked as death. When relevant, mice were treated with blocking (anti-mouse NKG2D HMGD2, BioXCell) or depleting antibodies (anti-mouse NK1.1 PK136 and anti-mouse CD3 17A2, both from BioXCell) injected intraperitoneally every 48 h starting 1 day prior to infection. Antibodies were diluted in sterile 1x PBS and injected in a final volume of 200 μl /mouse. For the fungal burden experiments, the mice were sacrificed 48 hours post-infection, and the kidneys and livers were harvested. The organs were homogenized manually, filtered via a 70 μm filter, serially diluted in 1x PBS, and plated on Sabouraud agar plates.

Following 48 h of incubation in a stationary 37 °C incubator, the fungal colonies that grew were counted.

For the immune cell profiling experiment, mice were sacrificed 48 hours post-infection, and their blood, kidneys, and liver were harvested. The organs were homogenized manually, filtered via a 70 μm filter, and resuspended in 1.5 ml 1x PBS followed by 3 repeats of a 2-min incubation in ACK medium (37 mM NH4Cl, 10 mM KHCO3, 0.1 M EDTA) at room temperature and centrifugation ($515 \times g$, 5 min). pellets were resuspended in sterile 1x PBS and analyzed by flow cytometry as described above.

**Murine vulvovaginal candidiasis model**
Female mice and *C. albicans* yeasts were grown as described. The mice were injected subcutaneously with 0.5 μg/200 μl of Estradiol solubilized in Sesame oil (both from Sigma-Aldrich) on days − 5, 1, and 8 prior to and following the fungal inoculation. On the day of inoculation, fungal cells were washed 3 times in sterile 1x PBS, counted using a hemocytometer, and diluted to a final concentration of $2.5 \times 10^7$ CFU/ml in sterile 1x PBS. The mice were then injected vaginally with $5 \times 10^5$ CFU/20μl/mouse. Two weeks post-infection, the mice were sacrificed, and vaginal lavage was performed using 100 μl sterile 1x PBS per mouse. The lavage was serially diluted, plated on Sabouraud agar plates, and incubated in a stationary 37 °C incubator. Two days later, fungal colonies were visualized and counted.

**Statistical analysis**
Statistical analysis was performed using either Prism 8 (GraphPad) or Excel (Microsoft). All the relevant statistical data for the experiments, including the statistical test used, the value of n, the definition of significance, etc., can be found in the figure legends or the relevant methods section.

**Reporting summary**
Further information on research design is available in the Nature Portfolio Reporting Summary linked to this article.

## Data availability
The data generated in this study are provided in the Supplementary Information/Source Data file. Source data are provided in this paper.

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

## Acknowledgements

Y.C.A. is supported by the Azrieli Foundation and the Foulkes Foundation. This work was supported by the following grants awarded to O.M.: the Israel Innovation Authority Kamin grant 62615, the German-Israeli Foundation for Scientific Research and Development grant 1412-414.13/2017, the ICRF professorship grant, the ISF Israel-China grant 2554/18, the MOST-DKFZ grant 3-14931, and the Ministry of Science and Technology grant 3-14764.

## Author contributions

Conceptualization, O.M. and Y.C.A.; Methodology, O.M., Y.C.A., M.K., N.O., and R.B.A.; Investigation, Y.C.A., M.K., R.K., B.I., T.B.B., and E.G.C.; Resources, A.A.K., R.B.A., M.K., N.G., and N.O.; Writing – Original Draft, O.M.,Y.C.A., and M.K.; Writing – Review & Editing, Y.C.A., M.K., R.K., B.I., T.B.B., A.A.K., R.B.A., N.O., and O.M.; Visualization, Y.C.A. and M.K.; Supervision, N.O. and O.M.; Project Administration, O.M.; Funding Acquisition, O.M. and Y.C.A.

## Competing interests

The authors declare no competing interests.
