## [Peer Review File · Nature Communications]

The activating receptor NKG2D is an anti-fungal pattern recognition receptorREVIEWER COMMENTS

Reviewer #1 (Remarks to the Author):

This manuscript by Charpak-Amikam et al. that describes for the first time that there are NKG2D ligands expressed on fungus is well-written and concise. The authors demonstrate that both mouse and human NKG2D binds to multiple fungi. Furthermore, they show that NK cells from both species affect fungi viability in vitro. They show both using mice genetically deficient in NKG2D and NKG2D blocking antibodies that NKG2D is critical for survival of mice infected with fungi. Further, they demonstrate this NKG2D-mediated protection requires both NK and T cells. These results are novel and of great interest with possible clinical relevance. In addition, the authors discuss the limitations of their study well.

I have only the following minor comments:

1. Validation of the NKG2D-Ig should be shown demonstrating it is NKG2D and not something else.
2. Fig. 1 and 2: Unstained cells should be shown in the histograms as well so it can be appreciated whether any of the staining from the Igs is non-specific.

Reviewer #2 (Remarks to the Author):

Charpak-Amikam et al. present an interest manuscript suggesting that the activating receptor NKG2D that is found on NK and CD8 T cells is an anti-fungal pattern recognition receptor. The premise of the paper is quite solid and exciting: the authors argue that C-type lectin family members, of which NKG2D is one, play a major role in the recognition of carbohydrates, and thus this could play a role in anti-fungal recognition. While several mammalian ligands for NKG2D are known and thought to be particularly critical in the recognition of tumor or virus-infected cells, the possibility that NKG2D could also serve to recognize fungal pathogens is an interesting new angle. While some of the data to support this idea are quite compelling, unfortunately, a lack of power, inadequate explanation of several key methods, lack of transparency into how some of the data were acquired, undermine the ability to support their conclusions. In its current form, the conclusions are not supported by rigorous investigation. Specific comments are below:

- A major issue throughout manuscript is the inappropriate use of statistics. A Student's T-test is inappropriate for sample sizes this small, and should only be used with normally distributed data. N=3 is simply inadequate to assess normality, and raises the concern that the authors chose this test because it would make their data significant with only a few data points. Moreover, although the samples in each group appear to be paired in many places, Student's T-test is used for unpaired data. A more appropriate test in this instance would be the Wilcoxon signed-rank test, which is a nonparametric test used on paired data. This concern affects virtually every figure and panel throughout the manuscript, and undermines confidence in the results.
- As a major (and exciting) finding of the manuscript is that NKG2D can bind to fungal cell walls, some demonstration that their engineered NKG2D-Fc is specific for NKG2D-ligands is needed. While I appreciate the inclusion of a control-Ig, it would be important to understand that the extracellular domain of NKG2D was intact and could bind normal partners after fused to the Fc. The authors should provide supplementary data to show reproducible binding to tumor cells known to express NKG2D with the absence of binding to primary cells were ligands are lowly and poorly expressed.
- The *C. albicans* data is generally convincing, but more is needed to demonstrate that NKG2D is also able to respond to *Cr. neoformans* and *A. fumigatus* as well as other species of *Candida*.
 - o The experiments shown in figure 1 should be replicated for *Cr. neoformans* and *A. fumigatus* to demonstrate binding of NKG2D to these organisms..
 - o The killing/degranulation experiments in Figure 2 should be replicated for *C. glabrata*, *C. parapsilosis*, and *C. tropicalis* to determine whether NK cells control the growth of these pathogens through NKG2D
- ♣ While the authors show that there is a technically statistically significant interaction between

NKG2D-Ig and *C. glabrata*, it is very unclear whether this seemingly small increase in binding would actually translate to relevant differences in the NK cell response. Showing data on the NK cell response to this stain would elucidate whether this interaction is biologically relevant.

- The paper would benefit from additional analysis into the phenotype of NK cells responding to these pathogens. For example, what is the phenotype of the degranulating (CD107a+) NK cells? How do different NK cell subsets (CD56bright, dim, etc) respond? Is there a correlation between expression of NKG2D on a donor's NK cells at baseline and their ability to control fungal replication?
- The paper would benefit from the presentation of the raw data on CFU counts, which could be included in a supplement. It is important for the reader to be able to evaluate the distribution of the data in terms of CFU counts and the impact of NK cells, and when this is collapsed into %CFU reduction this is difficult to assess (and such division can amplify error). With this in mind, it is also important to understand the reproducibility of the findings. In Figure 2BC, each line represents a single NK cell donor against a set amount of fungus (presumably). It would be helpful to see replicates within the same donor—if the same NK cells are used would the same % reduction be observed. This is critical data to understand the rigor and reliability, and should be provided.
- Figs. 2F, 2G, and 2I seem to show that *Cr. neoformans* have enhanced growth in the presence of NK cells from some donors. Their growth is further increased in the presence of NKG2D blocking Ab, which is in line with their stated hypothesis, but the fact that NK cells from some donors seem to enhance fungal growth undermines their point that NK cells are important in control of these pathogens.
- Given the huge amount of heterogeneity observed in the responses, the authors should repeat the experiments in figure 2 with more healthy donors and appropriate statistical tests. It would also be ideal to use the same donors across all experiments to observe whether NK cells from the same donor respond differently to different fungal pathogens.
- The study would hugely benefit from additional mechanistic work further exploring the recognition of fungal pathogens by NKG2D. In the authors' previous manuscript, <https://www.ncbi.nlm.nih.gov/pmc/articles/PMC9072312/>, they explore the interactions between various *C. albicans* proteins (notably, the Als family) and TIGIT and are able to identify the specific proteins recognized by TIGIT. The authors should perform similar experiments to identify the specific molecules interacting with NKG2D. These experiments would not only further inform the interaction between NKG2D and *C. albicans*, but would support (or refute) potential interactions with other fungal pathogens, given that other pathogens may or may not share the relevant molecules interacting with NKG2D. While we recognize that carbohydrate antigens can be more difficult to characterize, this is not impossible. In fact, an experiment to remove the sugars on fungi to see if the NK cells lose recognition seems a necessary experiment to have confidence in their model. This is feasible likely through chitinases, chitobioses and alpha and beta-D-Glucanases.
- The methods state that "The [NK] cells were co-cultured in 96 well U bottomed plates in the presence of irradiated RPMI8866 cells (5×10^3 /well) and irradiated PBMCs from two independent donors (5×10^4 /well per donor)". Why was this done? For how long was this co-culture performed? It is absolutely unclear why the NK cells would be pre-activated in this way and is not in line with their premise. As NKG2D is expressed at baseline on virtually 100% of NK cells, and if it is playing a role in NK cell recognition, then such pre-activation should not be necessary. Either way, this can influence the results and should be addressed.
- The authors do not describe the methods for their degranulation assay. What was the effector to target (E:T) ratio used in this assay? Were the NK cells freshly isolated or used after expansion with feeder cells? For how long did the co-culture take place? The timing of CD107a antibody inclusion should also be added
- Clarification of their control Ig (and what it targets) would be helpful.
- Some demonstration (in a supplement) that the metabolic activity a good measure of the number of fungi present for 2I is needed.
- Given that the in vivo experiments showed that CD8 was critical as well, it would greatly benefit the story to also evaluate CD8 T cell responses in vitro to these fungi
- OK to include in the supplement, but it would be helpful to remind the reader the characteristics of the NKG2D KO mice. Are these mice able to kill other types of targets?
- The authors should quantify NK and CD8 T cells in knockouts and normal, and the in the setting of liver and kidney infection.

Reviewer #3 (Remarks to the Author):

The authors use an unbiased approach to generate NK receptor - Ig fusion proteins and test which constructs bind to *Candida albicans*. The authors identified a fusion protein containing NKG2D. They show the construct binds to other *Candida* species and *C. neoformans* and interfere with NK killing of *Candida*, *Cryptococcus* and possibly *Aspergillus*. They go on to show that NKG2D is necessary for optimal host defense in a mouse model of candidiasis.

Major Comments

1. The authors show NKG2D binds to fungi and is necessary for NK cell killing and host defense. NKG2D could function as an activating receptor and signal for fungal cytotoxicity. Alternately, NKG2D could work solely as an adhesion molecule required for another NK-activating receptor to mediate cytotoxicity. The authors should distinguish between these possibilities.
2. The authors use an unbiased approach to identify NKG2D. The manuscript would be stronger if they were to show some of the fusion constructs that didn't work, highlighting a strength of the manuscript.

Minor Comment

1. Previous studies have shown NK cells bind and kill hyphae of *Aspergillus* rather than conidia. Do the authors have information on the morphology that NKG2D recognizes? This might provide important information about the ligand.
2. The binding of the fusion protein to *C. glabrata* was similar to *C. krusei*. Although the former reached statistical significance and the latter did not, the authors may wish to state if either is likely to be biologically important.
3. Did the NKG2D fusion protein bind to the blocking antibody?
4. Line 153, the authors state, "NK cells were able to reduce the number of viable *A. fumigatus* cells, but only assess metabolic activity and do not assess death. They should use more precise wording.

Reviewer #1 (Remarks to the Author):

This manuscript by Charpak-Amikam et al. that describes for the first time that there are NKG2D ligands expressed on fungus is well-written and concise. The authors demonstrate that both mouse and human NKG2D binds to multiple fungi. Furthermore, they show that NK cells from both species affect fungi viability in vitro. They show both using mice genetically deficient in NKG2D and NKG2D blocking antibodies that NKG2D is critical for survival of mice infected with fungi. Further, they demonstrate this NKG2D-mediated protection requires both NK and T cells. These results are novel and of great interest with possible clinical relevance. In addition, the authors discuss the limitations of their study well.

We thank reviewer #1 for the positive review and the interesting comments. We have amended the manuscript in accordance with the following comments:

I have only the following minor comments:

1. Validation of the NKG2D-Ig should be shown demonstrating it is NKG2D and not something else.

The reviewer raises a critical issue, as many of the findings are indeed based on NKG2D-Ig. As such we routinely test NKG2D-Ig before using it. We did not specify these internal quality control steps in the original text, but we now understand they can be of value and we have added the following:

- A. An agarose gel in which we run denatured and non-denatured NKG2D-Ig followed by Coomassie staining (supplemental fig. 3A). The experiment shows that NKG2D-Ig is at the expected size (and runs as a dimer in its non-denatured state). Under reducing conditions two protein bands are seen that probably reflects different glycosylation of NKG2D-Ig. This two bands appearance is observed in many fusion proteins that we are working with, for example: NKp46-Ig, NKp44-Ig NKp30-Ig and more.
- B. To validate the functionality & specificity of our NKG2D-Ig proteins we used flow cytometry to stain two cell lines. Vero, which is a monkey cell line known to express only negligible levels of NKG2D ligands, and Vero-

MICA, in which we artificially expressed the NKG2D ligand MICA. As can be seen (Supplemental figure 3B), only the MICA-expressing cells were stained by NKG2D-Ig.

- C. Finally, to re-validate the specificity of NKG2D-Ig we coated ELISA plates with various NK receptor fusion proteins and stained them with several antibodies (Supplemental figure 3C). As expected, each fusion protein was only stained by its specific antibody.

2. Fig. 1 and 2: Unstained cells should be shown in the histograms as well so it can be appreciated whether any of the staining from the Igs is non-specific.

In our experience, some of the Ig-fusion proteins have a non-specific binding. Due to that, instead of using unstained cells we use cells stained with an unrelated fusion protein (usually NKp46-Ig or its derivative NKp46-D1-Ig in the experiments presented in this manuscript, as specified in the methods section). We only consider the binding reliable and the results positive if the tested fusion protein (NKG2D-Ig in this case) doesn't only bind significantly more than unstained cells (which it will many times due to the non-specific binding tendency of fusion proteins), but also more than staining with other fusion proteins (which share this baseline non-specific binding).

Reviewer #2 (Remarks to the Author):

Charpak-Amikam et al. present an interest manuscript suggesting that the activating receptor NKG2D that is found on NK and CD8 T cells is an anti-fungal pattern recognition receptor. The premise of the paper is quite solid and exciting: the authors argue that C-type lectin family members, of which NKG2D is one, play a major role in the recognition of carbohydrates, and thus this could play a role in anti-fungal recognition. While several mammalian ligands for NKG2D are known and thought to be particularly critical in the recognition of tumor or virus-infected cells, the possibility that NKG2D could also serve to recognize fungal pathogens is an interesting new angle. While some of the data to support this idea are quite compelling, unfortunately, a lack of power, inadequate explanation of several key methods, lack

of transparency into how some of the data were acquired, undermine the ability to support their conclusions. In its current form, the conclusions are not supported by rigorous investigation. Specific comments are below:

- A major issue throughout manuscript is the inappropriate use of statistics. A Student's T-test is inappropriate for sample sizes this small, and should only be used with normally distributed data. N=3 is simply inadequate to assess normality, and raises the concern that the authors chose this test because it would make their data significant with only a few data points. Moreover, although the samples in each group appear to be paired in many places, Student's T-test is used for unpaired data. A more appropriate test in this instance would be the Wilcoxon signed-rank test, which is a nonparametric test used on paired data. This concern affects virtually every figure and panel throughout the manuscript, and undermines confidence in the results.

We understand the reviewer's concern about the choice of statistical tests. We have re-analysed our data using appropriate tests. As suggested, we mainly used the nonparametric Wilcoxon signed-rank test, or Wilcoxon matched-pairs signed rank test where appropriate. As in these tests the minimal number of repeats necessary to reach significance is N=5, we had to add repeats to many of the experiments (Fig. 1A+D+G, Fig. 2E+G+H+J+M, Fig. 4G). We are happy to report that while strengthening the statistical validity of our findings, these changes did not change any of our original findings.

In addition, where appropriate and as accepted in the field we also used a Mann-Whitney test for some of the in-vivo fungal burden experiments (Fig. 3F, 4A-C) Log-rank (Mantel-Cox) test for in-vivo survival results (Fig. 3A+C) or 2-way ANOVA (4D-F).

- As a major (and exciting) finding of the manuscript is that NKG2D can bind to fungal cell walls, some demonstration that their engineered NKG2D-Fc is specific for NKG2D-ligands is needed. While I appreciate the inclusion of a control-Ig, it would be important to understand that the extracellular domain of NKG2D was intact and could bind normal partners after fused to the Fc. The authors should provide

supplementary data to show reproducible binding to tumor cells known to express NKG2D with the absence of binding to primary cells where ligands are lowly and poorly expressed.

We agree with the critical point that reviewer #2 raised here, that we need to better describe the quality control measure we routinely use to validate the identity and integrity of our Ig fusion proteins.

Our quality control experiments for the Ig fusion proteins, that we now present as supplementary figure 3, are:

- A. An agarose gel in which we run denatured and non-denatured NKG2D-Ig followed by Coomassie staining (supplemental fig. 3A). The experiment shows that NKG2D-Ig is at the expected size (and runs as a dimer in its non-denatured state). Under reducing conditions two protein bands are seen that probably reflect different glycosylation of NKG2D-Ig. This two bands appearance is observed in many fusion proteins that we are working with, for example: NKp46-Ig, NKp44-Ig, NKp30-Ig and more.
- B. To validate the functionality & specificity of our NKG2D-Ig proteins we used flow cytometry to stain two cell lines. Vero, which is a monkey cell line known to express only negligible levels of NKG2D ligands, and Vero-MICA, in which we artificially expressed the NKG2D ligand MICA. As can be seen (Supplemental figure 3B), only the MICA-expressing cells were stained by NKG2D-Ig.
- C. Finally, to re-validate the specificity of NKG2D-Ig we coated ELISA plates with various NK receptor fusion proteins and stained them with several antibodies (Supplemental figure 3C). As expected, each fusion protein was only stained by its specific antibody.

- The *C. albicans* data is generally convincing, but more is needed to demonstrate that NKG2D is also able to respond to *Cr. neoformans* and *A. fumigatus* as well as other species of *Candida*.

- o The experiments shown in figure 1 should be replicated for *Cr. neoformans* and *A. fumigatus* to demonstrate binding of NKG2D to these organisms..

We thank the reviewer for this comment and agree that in order to substantiate our claims we need to not only show the functional effects of NKG2D blockade on cytotoxicity, but also that NKG2D directly binds these organisms.

1. Regarding *A. fumigatus*: it is technically problematic to use FACS to stain this fungus, as it is a mold. As a mold, it usually grows as large biofilms that are both challenging to stain uniformly and be stained using flow cytometer. Due to that we decided to test direct NKG2D binding to *A. fumigatus* using an ELISA assay. In this assay we grew *A. fumigatus* cells in ELISA plate wells and then stained them using NKG2D-Ig (as described in detail in the revised methods section). As can be seen in the new Fig. 2K, we observed direct binding of NKG2D to *A. fumigatus* cells.

2. Regarding *Cr. Neoformans* – due to technical limitations of observing the cells of this organism using several flow cytometers we have in our possession we decided to perform an ELISA assay to show direct binding of NKG2D to *Cr. Neoformans*. These results appear now as fig. 2G+H and indeed show direct binding to this fungus as well.

o The killing/degranulation experiments in Figure 2 should be replicated for *C. glabrata*, *C. parapsilosis*, and *C. tropicalis* to determine whether NK cells control the growth of these pathogens through NKG2D

The initial idea behind the experiment now shown in fig. 2A-E (NKG2D-Ig staining of non-*albicans* *Candida* species) was to check the specificity of our findings and to get a basic idea of the possible target range of NKG2D. Once we identified binding to *Candida* species other than *C. albicans* we moved on to fungal pathogens that are further away evolutionary, structurally and functionally: *Cryptococcus neoformans* and *Aspergillus fumigatus*. Once we noticed that these species are also bound by NKG2D, we decided to focus on them, assuming that whatever effect is true for both *C. albicans*, *C. neoformans* and *A. fumigatus* is most likely also shared by closer relatives such as those presented in fig 2A-E.

Performing a cytotoxicity assay demands rigorous calibration for each organism (the conditions needed by *Candida albicans* are very different than the ones needed by *Candida glabrata*, those needed by *C. parapsilosis*, etc.). This is why we now focused on the species that are farthest from each other and are most relevant clinically (*C. albicans*, *C. neoformans* and *A. fumigatus*).

Due to the abovementioned time constrains we invested our time during these

corrections in showing cytotoxicity against only one non-albicans Candida species: *C. glabrata*. This species was chosen due to its being the most different from *C. albicans* (by both evolutionary distance and pathogenesis) and the 2nd most common Candida-family pathogen after *C. albicans*. As can be seen in the new fig. 2F, blocking NKG2D impairs NK cytotoxicity vs. this fungus as well, similar to all other fungal pathogens tested.

We assume this to be true for most other fungal species in the evolutionary space between *Candida*, *Cryptococcus* and *Aspergillus*, including the untested fungi shown in fig. 2A-E. In case the reviewer does not agree with this assumption and feels that our conclusions are not validated enough, we suggest removing this data from the text altogether (as the experiments done on *C. neoformans*, *A. fumigatus* and *C. glabrata* are enough for us to make our major point: that NKG2D binds and mediates immune responses to very diverse fungal pathogens).

♣ While the authors show that there is a technically statistically significant interaction between NKG2D-Ig and *C. glabrata*, it is very unclear whether this seemingly small increase in binding would actually translate to relevant differences in the NK cell response. Showing data on the NK cell response to this stain would elucidate whether this interaction is biologically relevant.

We agree with the reviewer's observation. We therefore performed a cytotoxicity assay of NK cells vs. *C. glabrata*, which is presented in the new fig. 2F. As can be seen in it, NKG2D blockade impairs the ability of NK cells to eliminate *C. glabrata* cells. As such, it seems that the weaker but statistically significant binding seen in our initial experiments indeed translates to a functional effect.

• The paper would benefit from additional analysis into the phenotype of NK cells responding to these pathogens. For example, what is the phenotype of the degranulating (CD107a+) NK cells? How do different NK cell subsets (CD56bright, dim, etc) respond? Is there a correlation between expression of NKG2D on a donor's NK cells at baseline and their ability to control fungal replication?

The reviewer's raises several interesting questions regarding our findings.

1. To further explore the phenotype of the NK cell subsets that react to fungi we

repeated our NK de-granulation experiments, this time also staining the cells for CD56 in order to differentiate between CD56^{dim} and CD56^{bright} cells. As our original protocol uses activated NK cells that are CD56^{dim}, now we used NK cells freshly isolated from human blood. As Can be seen in the new fig. 1H, both the CD56^{dim} and the CD56^{bright} populations de-granulate in response to *Candida albicans*.

2. We also checked whether there is a correlation between NKG2D levels on a donor's NK cells and their degranulation levels following recognition of *C. albicans*. To that end we repeated the above-mentioned experiment but also stained for NKG2D. We then calculated the R² coefficient of determination values of these double-stainings in both the CD56^{dim} and CD56^{bright} populations. As presented in fig.1I, no correlation was observed.

- The paper would benefit from the presentation of the raw data on CFU counts, which could be included in a supplement. It is important for the reader to be able to evaluate the distribution of the data in terms of CFU counts and the impact of NK cells, and when this is collapsed into %CFU reduction this is difficult to assess (and such division can amplify error). With this in mind, it is also important to understand the reproducibility of the findings. In Figure 2BC, each line represents a single NK cell donor against a set amount of fungus (presumably). It would be helpful to see replicates within the same donor—if the same NK cells are used would the same % reduction be observed. This is critical data to understand the rigor and reliability, and should be provided.

1. We show absolute CFU values for the in-vivo experiments as they are indeed important due to all the reasons the reviewer mentions.

For the in-vitro experiments we show relative values to better compare between different experiments. We always start from the same number of fungal cells (1000/well), but each experiment takes many hours, in which the fungal cells grow logarithmically in parallel to the elimination of a certain percent of them by the NK cells. During this time, a second process takes place: the presence of mammalian cells accelerates growth of *Candida* cells, probably because fungal cells kill mammalian cells (including immune cells) and use them as a food source. The killing assay is actually a complex competition in which NK cells kill some of the fungal

cells, and fungal cells kill some of the NK cells. We measure how many fungal cells were eliminated by the NK cells, but different fungal growth rates due to small differences in the media (different batches of Fetal calf serum, etc.) and biological difference between NK cell lines from different donors lead to different CFU values that can be of different orders of magnitude. We therefore present this data as %CFU reduction to be able to present it coherently to the reader, and in order to be able to compare between experiments.

It is important to mention that the above effect is due to the in-vitro conditions favoring the microbial cells. In vivo, in the presence of all the immune-system's redundancies (fever, additional immune cells, extreme numerical advantage for the immune cells, cytokines that enhance NK cell activity, etc.), NK cells are much more important and efficient against fungi (as shown in a previous paper of ours where we depleted NK cells during in-vivo *Candida* infection – Charpak-Amikam et. al., *Nat. Commun.*, 2022).

2. We completely agree with the reviewer's suggestion about the reproducibility of our findings and performing technical repeats. All fungal killing assays are repeated as the reviewer suggests, and we repeat each donor-fungus combinations several times due to the reasons the reviewer mentions. We present the data as a single point per donor per condition (which is the average of the technical repeats) as otherwise it makes the figure very crowded and much harder to decipher and understand. We of course use both the average and the raw values in our analysis and our quality control of the data, and we don't use experiments in which the technical repeats point to a technical problem, and their average does not seem to represent the real result.

We now better explain this in the relevant figure legends, please see figures 1F-H, 2F+I+J. In addition, we attach here a non-normalized anti-fungal cytotoxicity results as the reviewer requested and in order to follow up on our above comments. For example, these results were normalized and joined to form figure 2F.

C. glabrata NK cytotoxicity experiment - technical repeats

• Figs. 2F, 2G, and 2I seem to show that *Cr. neoformans* have enhanced growth in the presence of NK cells from some donors. Their growth is further increased in the presence of NKG2D blocking Ab, which is in line with their stated hypothesis, but the fact that NK cells from some donors seem to enhance fungal growth undermines their point that NK cells are important in control of these pathogens.

This is a true and interesting observation made by the reviewer. This has been observed by us since we began working on NK-fungal interactions. The reasons behind this effect are elaborated on in our answer to the previous comment.

• Given the huge amount of heterogeneity observed in the responses, the authors should repeat the experiments in figure 2 with more healthy donors and appropriate statistical tests. It would also be ideal to use the same donors across all experiments to observe whether NK cells from the same donor respond differently to different fungal pathogens.

1. Regarding the number of repeats – as the reviewer suggested above, we repeated most of the experiments described in the manuscript and changed the statistical tests to ones that are indeed more appropriate. This is explained in detail in our response to the reviewer's first comment.

2. Regarding the usage of the same donors vs different fungi: We thank the reviewer as this is a very interesting idea. Unfortunately, this is nearly impossible technically due to the short lifespan of each line ex-vivo, and due to the random and double-blinded nature of our donors (we usually use blood donations from anonymous

donors) which prevents us from sampling blood from the same donors multiple times.

- The study would hugely benefit from additional mechanistic work further exploring the recognition of fungal pathogens by NKG2D. In the authors' previous manuscript, <https://www.ncbi.nlm.nih.gov/pmc/articles/PMC9072312/>, they explore the interactions between various *C. albicans* proteins (notably, the Als family) and TIGIT and are able to identify the specific proteins recognized by TIGIT. The authors should perform similar experiments to identify the specific molecules interacting with NKG2D. These experiments would not only further inform the interaction between NKG2D and *C. albicans*, but would support (or refute) potential interactions with other fungal pathogens, given that other pathogens may or may not share the relevant molecules interacting with NKG2D. While we recognize that carbohydrate antigens can be more difficult to characterize, this is not impossible. In fact, an experiment to remove the sugars on fungi to see if the NK cells lose recognition seems a necessary experiment to have confidence in their model. This is feasible likely through chitinase, chitinases and alpha and beta-D-Glucanases.

We thank the reviewer for this comment and agree that identification of the fungal ligand or ligands that NKG2D recognizes is one of the most interesting questions that remains open in our study.

To attempt and answer this question we decided to not test potential carbohydrates one by one enzymatically, but to perform a high-throughput screen. We collaborated with a group in the Imperial College of London and performed binding assays with NKG2D-Ig against an array containing hundreds of carbohydrates from various sources including many fungi, bacteria and plants. This data is now presented in supplementary figure 2 and supplementary data 1. Unfortunately, no significant binding was detected.

Therefore, we now have less confidence in our carbohydrate ligand hypothesis, and we have updated the text appropriately. We understand that the nature of the ligand/ligands and its identification is, while fascinating, a long and complicated process. As the reviewer knows it is very difficult to identify ligands for NK cell receptors. For example, not all tumor ligands for NKp46 were identified till today although NKp46 was identified around 30 years ago and is one of the more central

and studied NK cell receptors. Thus, finding the fungal ligand (if ever successful) will probably be a part of a future manuscript.

- The methods state that “The [NK] cells were co-cultured in 96 well U bottomed plates in the presence of irradiated RPMI8866 cells (5×10^3 /well) and irradiated PBMCs from two independent donors (5×10^4 /well per donor)”. Why was this done? For how long was this co-culture performed? It is absolutely unclear why the NK cells would be pre-activated in this way and is not in line with their premise. As NKG2D is expressed at baseline on virtually 100% of NK cells, and if it is playing a role in NK cell recognition, then such pre-activation should not be necessary. Either way, this can influence the results and should be addressed.

The presence of the feeder cells is indeed needed for NK cell activation ex-vivo and for their maintenance for a long period of time (around one month). These stimulating conditions include many NK ligands that are presented on the “feeder” cells, and various cytokines and other signaling molecules. We use these conditions as these activation conditions allow our human NK lines to survive and be active for several weeks. This greatly reduces our demand of additional blood donations, and we consider this both more moral and more efficient.

To further explore the interaction between NK cells and fungi we now performed experiments with freshly isolated NK cells. This is the set of experiments with CD56^{dim/bright} cells we described in an earlier comment, which are now presented in fig. 1H+I. In addition, the splenic mouse NK cells used in Fig. 4G (cytotoxicity vs. *C. albicans* with NK cells from WT and NKG2D KO mice) are also fresh and did not undergo ex-vivo activation or a long period ex-vivo prior to the experiment.

- The authors do not describe the methods for their degranulation assay. What was the effector to target (E:T) ratio used in this assay? Were the NK cells freshly isolated or used after expansion with feeder cells? For how long did the co-culture take place? The timing of CD107a antibody inclusion should also be added

We thank the reviewer for noticing our oversight. We have now added this to the methods section.

- Clarification of their control Ig (and what it targets) would be helpful.

This is indeed a critical detail as this control is in many of the manuscript's central experiments. Its identity is detailed in the methods sections of both the flow cytometry, the Fc fusion protein generation, and the ELISA experiments.

- Some demonstration (in a supplement) that the metabolic activity a good measure of the number of fungi present for 2I is needed.

We understand the reviewer's concern about the XTT metabolic activity method, as it is indeed very different than the other methods we use to quantify fungi in most of our experiments (mainly CFU). As *A. Fumigatus* is a mold, it is technically very challenging to grow and quantify as colonies following co-culture with NK cells for several hours. As such, we chose to use a different method. After reading the literature and consulting with experts in the field of *Aspergillus* biology we decided on using the XTT assay for metabolic activity. This was chosen as this is a well-established and widely used method in the field for dozens of years. It has been used and tested in many independent laboratories and projects, and we add here a short list of examples from various publications that calibrated it and used it:

1. Meletiadis et. al., Colorimetric assay for antifungal susceptibility testing of *Aspergillus* species, *J Clin. Microbiol.*, 2001.
2. Ramage et. al., Standardized method for in vitro antifungal susceptibility testing of *Candida albicans* biofilms, *Antimicrob Agents Chemother.*, 2001
3. Nett et. al., Optimizing a *Candida* biofilm microtiter plate model for measurement of antifungal susceptibility by tetrazolium salt assay, *J Clin. Microbiol.*, 2011.
4. Ramirez-Ortiz et. al., A nonredundant role for plasmacytoid dendritic cells in host defense against the human fungal pathogen *Aspergillus fumigatus*, *Cell Host Microbe*, 2011.
5. S
a

- Given that the in vivo experiments showed that CD8 was critical as well, it would greatly benefit the story to also evaluate CD8 T cell responses in vitro to these fungi
We agree with the reviewer that the NKG2D-mediated T cell recognition of fungal cells is indeed an interesting and important question. As the killing of CD8+ T cells require MHC restriction and MHC molecules are not present on fungi cells, the way

e

t

.

by which CD8+ T cells recognize and kill fungi is unknown and we are unfamiliar with proper methods by which to study this subject in vitro. As such, the best tool we have for this question are in-vivo models, similar to the ones used in fig. 4C.

If the reviewer is familiar with any appropriate in-vitro experiment to approach this question we would be happy to perform it. As this is indeed an important point, we now also discuss it in the discussion.

- OK to include in the supplement, but it would be helpful to remind the reader the characteristics of the NKG2D KO mice. Are these mice able to kill other types of targets?

We agree with the reviewer that this strain of mice is central in our experiments, and we should further elaborate on its characteristics. As such, we have amended the relevant paragraph in the results section appropriately.

- The authors should quantify NK and CD8 T cells in knockouts and normal, and the in the setting of liver and kidney infection.

We agree with the reviewer that this is an important control in order to make sure that the in vivo effect we see is due to different activity of NKG2D in the two mice strains, and not a developmental difference in the quantity and/or localization of NK and T cells in the tissues relevant to our model.

To that end we have repeated our in-vivo model in NKG2D^{+/+} and NKG2D^{-/-} mice, extracted blood, kidneys and livers from them, and used flow cytometry to quantify the relevant immune cell populations. The results are now presented as fig. 4D-F and show no significant reduction in NK, CD8 T cell or CD4 T cell numbers in these tissues in KO mice under basal conditions. On the contrary - we found a general increase in NK cell numbers in the kidneys before and after infection (relative to WT mice), and an increase in basal kidney CD8 T cell numbers that normalizes to WT levels following infection. In the livers of KO mice we found no changes in NK cell numbers, and no change in basal CD8 T cell numbers, but we did observe a reduction in CD8 T cell numbers after infection.

Reviewer #3 (Remarks to the Author):

The authors use an unbiased approach to generate NK receptor - Ig fusion proteins and test which constructs bind to *Candida albicans*. The authors identified a fusion protein containing NKG2D. They show the construct binds to other *Candida* species and *C. neoformans* and interfere with NK killing of *Candida*, *Cryptococcus* and possibly *Aspergillus*. They go on to show that NKG2D is necessary for optimal host defense in a mouse model of candidiasis.

We thank reviewer #3 for the review and his thoughtful comments. We have amended the manuscript in accordance and as specified below:

Major Comments

1. The authors show NKG2D binds to fungi and is necessary for NK cell killing and host defense. NKG2D could function as an activating receptor and signal for fungal cytotoxicity. Alternately, NKG2D could work solely as an adhesion molecule required for another NK-activating receptor to mediate cytotoxicity. The authors should distinguish between these possibilities.

The reviewer raises an important and interesting question regarding the mechanism of action by which NKG2D leads to immune cell activation following fungal recognition. To differentiate between direct NKG2D signalling and adhesion we performed a set of co-immunoprecipitation experiments. In these experiments we co-cultured NK and *C. albicans* cells (or, as control, NK cells alone), prepared lysates of the NK cells and then immunoprecipitated DAP10. We then performed a western blot using antibodies against tyrosine phosphate and against DAP10. This blot was used to measure whether the presence of fungal cells indeed leads to NKG2D signalling. DAP10 was used as it is the sole signalling mediator of NKG2D in humans and as no good immunoprecipitating anti-NKG2D antibodies are currently available (to the best of our knowledge).

As presented in fig. 1J, this experiment clearly shows that NKG2D recognition of fungal cells leads to DAP10 phosphorylation and downstream signalling. This supports the 1st hypothesis the reviewer suggested, that NKG2D functions as an activating receptor and not solely as an adhesion molecule.

2. The authors use an unbiased approach to identify NKG2D. The manuscript would be stronger if they were to show some of the fusion constructs that didn't work, highlighting a strength of the manuscript.

We agree with the reviewer's comment and as such we originally included negative Ig controls. The "control-Ig" presented in our manuscript is NKp46-Ig and its derivative D1-Ig, as is written in the relevant methods section. In addition, and as the reviewer suggested we have added another such negative result to fig. 1B: The NK Fc-receptor CD16.

Minor Comment

1. Previous studies have shown NK cells bind and kill hyphae of *Aspergillus* rather than conidia. Do the authors have information on the morphology that NKG2D recognizes? This might provide important information about the ligand.

The reviewer raises an interesting question regarding the specific morphological form of *A. fumigatus* that NKG2D recognizes. This is indeed interesting as the different morphologies have different roles in different stages of infection and as the reviewer mentions, different immune recognition/evasion patterns.

In both experiments in which we test the interaction between NKG2D and *A. fumigatus* we use a mixed population of fungal cells, where both morphologies (conidia and hyphae) are present. Luckily, in the XTT metabolic activity assay (fig. 2M) the main form is the hyphae, while in the ELISA experiment (fig. 2K) the main form is conidia. As such, our results show that both conidia and hyphae forms are recognized by NKG2D, but the specific contribution of each form is still unknown to us as in both experiments the population is dominated by one form, but still contains both. We now discuss this in the discussion section.

2. The binding of the fusion protein to *C. glabrata* was similar to *C. krusei*. Although the former reached statistical significance and the latter did not, the authors may wish to state if either is likely to be biologically important.

We agree that the binding results presented in fig. 2E themselves are not enough to conclude functionality of NKG2D. As such we have both added more repeats to this

experiment and have performed a cytotoxicity experiment to validate the effect of *C. glabrata* recognition by NKG2D (fig. 2F). As can be now seen, the binding we detected by flow cytometry is indeed also translated to function and NKG2D blockade impairs the ability of NK cells to recognize and eliminate *C. glabrata* cells.

3. Did the NKG2D fusion protein bind to the blocking antibody?

Yes, the antibody clone we used for blocking NKG2D can bind the NKG2D-Ig. They are only used in the same experiment in the ELISA assay to validate the identity of the Ig fusion proteins (supplemental fig. 3C), where we use this blocking antibody to recognize NKG2D-Ig.

4. Line 153, the authors state, "NK cells were able to reduce the number of viable *A. fumigatus* cells, but only assess metabolic activity and do not assess death. They should use more precise wording.

This is absolutely true and we thank the reviewer for finding this error. The relevant text has been amended.

REVIEWERS' COMMENTS

Reviewer #1 (Remarks to the Author):

All of my concerns have been sufficiently addressed.

Reviewer #2 (Remarks to the Author):

Overall, the paper has been extensively revised and greatly improved, addressing most of my major concerns.

One aspect of the rebuttal is lacking with regard to the role of CD8 T cells in this recognition mechanism. The authors make the assumption that the only way T cells could function (and kill) would be through TCR/MHC interactions. Simply put, this is a false assumption. It is possible that T cells are responding to some combination of NKG2D ligation plus other ligand or cytokine signals in the overall environment to drive TCR-independent T cell activation. Further, CD8 T cells could be secreting cytokines that inhibit fungal growth; this also could occur in a TCR independent fashion as bystander activation of T cells can occur. To study this in vitro would simply involve similar experiments as they performed with NK cells. Further, their statement in the first paragraph of the discussion is inaccurate because of this. They state as fact that MHC restriction is required. However, there is abundant evidence that bystander activation of CD8 T cells can occur (for instance, <https://www.nature.com/articles/s12276-019-0316-1>, but many more articles) and such mechanisms could similarly play a role in fungal infection. This assumption should be reconsidered, and the statements about MHC restriction on line 313 should be revised.

Reviewer #3 (Remarks to the Author):

The authors have addressed my comments, and I have no other concerns. However, the authors may wish to consider the comments below, which may enhance the presentation.

Line 119: The statement "NK cells directly kill their targets by the release of CD107a+ granules" is a little misleading. The authors imply they measured CD107a on released granules. Rather, they measured the CD107 on the NK cells after the granules fused with the NK cell membrane released their contents, which exposed CD107a. It would help to clarify this point.

The figures showing the reduction in CFU are confusing (Figure 1A, 2F, etc.). The reader has to understand that a negative reduction in CFU is actually growth. Thus, a ~-100% reduction means there are more organisms than a ~-25% reduction in CFU (Figure 2F). Surely there is a more intuitive way to express this data such as Figure 3F. At the very least, a label showing a downward arrow labelled "growth" would be helpful.

Readers will want to know why the authors think the following observations are "interesting". (lines 283-284), "Interestingly, in the main site of infection, the kidneys, we observed a significant increase in the basal levels all three lymphocyte populations in KO mice" and (lines 286-287), "Although we observed interesting changes in lymphocyte immune populations during infection of NKG2D KO mice with *C. albicans*".

REVIEWERS' COMMENTS

Reviewer #1 (Remarks to the Author):

All of my concerns have been sufficiently addressed.

We thank the reviewer for the valuable and insightful comments during the previous review process, and for confirming that all concerns have been thoroughly addressed.

Reviewer #2 (Remarks to the Author):

Overall, the paper has been extensively revised and greatly improved, addressing most of my major concerns.

One aspect of the rebuttal is lacking with regard to the role of CD8 T cells in this recognition mechanism. The authors make the assumption that the only way T cells could function (and kill) would be through TCR/MHC interactions. Simply put, this is a false assumption. It is possible that T cells are responding to some combination of NKG2D ligation plus other ligand or cytokine signals in the overall environment to drive TCR-independent T cell activation. Further, CD8 T cells could be secreting cytokines that inhibit fungal growth; this also could occur in a TCR independent fashion as bystander activation of T cells can occur. To study this in vitro would simply involve similar experiments as they performed with NK cells. Further, their statement in the first paragraph of the discussion is inaccurate because of this. They state as fact that MHC restriction is required. However, there is abundant evidence that bystander activation of CD8 T cells can occur (for instance, <https://www.nature.com/articles/s12276-019-0316-1>, but many more articles) and such mechanisms could similarly play a role in fungal infection. This assumption should be reconsidered, and the statements about MHC restriction on line 313 should be revised.

We thank the reviewer for the previous comments and for the insightful correction provided. We acknowledge the possibility that bystander T cells may indeed play a role in fungal elimination via NKG2D activation, and we have revised our discussion to reflect this suggestion. We will investigate this intriguing topic in our future studies.

Reviewer #3 (Remarks to the Author):

The authors have addressed my comments, and I have no other concerns. However, the authors may wish to consider the comments below, which may enhance the presentation.

Line 119: The statement “NK cells directly kill their targets by the release of CD107a+ granules” is a little misleading. The authors imply they measured CD107a on released granules. Rather, they measured the CD107 on the NK cells after the granules fused with the NK cell membrane released their contents, which exposed CD107a. It would help to clarify this point.

The figures showing the reduction in CFU are confusing (Figure 1A, 2F, etc.). The reader has to understand that a negative reduction in CFU is actually growth. Thus, a ~-100% reduction means there are more organisms than a ~-25% reduction in CFU (Figure 2F). Surely there is a more intuitive way to express this data such as Figure 3F. At the very least, a label showing a downward arrow labelled “growth” would be helpful.

Readers will want to know why the authors think the following observations are “interesting”. (lines 283-284), “Interestingly, in the main site of infection, the kidneys, we observed a significant increase in the basal levels all three lymphocyte populations in KO mice” and (lines 286-287), “Although we observed interesting changes in lymphocyte immune populations during infection of NKG2D KO mice with *C. albicans*”.

We thank the reviewer for the previous comments and for the additional feedback provided. We have revised the relevant sections accordingly.